# INTENTIONAL-GESTURE: DELIVER YOUR INTENTIONS WITH GESTURES FOR SPEECH

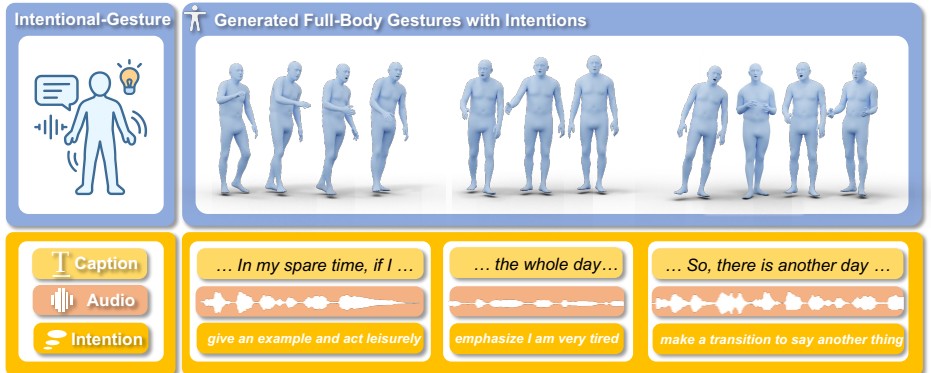

Figure 1: We present *Intentional Gesture*, a novel framework for intention-controllable gesture generation. Our method models latent communicative functions from speech and grounds motion generation in these inferred intentions.

## ABSTRACT

When humans speak, gestures help convey communicative intentions, such as adding emphasis or describing concepts. However, current co-speech gesture generation methods rely solely on superficial linguistic cues (*e.g.* speech audio or text transcripts), neglecting to understand and leverage the communicative intention that underpins human gestures. This results in outputs that are rhythmically synchronized with speech but are semantically shallow. To address this gap, we introduce **Intentional-Gesture**, a novel framework that casts gesture generation as an intention-reasoning task grounded in high-level communicative functions. First, we curate the **InG** dataset by augmenting BEAT-2 with gesture-intention annotations (*i.e.*, text sentences summarizing intentions), which are automatically annotated using large vision-language models. Next, we introduce the **Intentional Gesture Motion Tokenizer** to leverage these intention annotations. It injects high-level communicative functions (*e.g.*, intentions) into tokenized motion representations to enable intention-aware gesture synthesis that are both temporally aligned and semantically meaningful, achieving new state-of-the-art performance on the BEAT-2 benchmark. Our framework offers a modular foundation for expressive gesture generation in digital humans and embodied AI.

## 1 INTRODUCTION

Gestures accompany speech to convey ideas and facilitate comprehension De Ruiter et al. (2012); Song et al. (2023); Burgoon et al. (1990), forming an essential part of human communication. As AI advances Lu et al. (2023; 2024b); Ren et al. (2025), enabling virtual avatars to produce expressive gestures will be critical for digital avatar construction Tang et al. (2025); Song et al. (2024c); Huang et al. (2024); Song et al. (2024a;b).

Recent works Liu et al. (2025b; 2023); Yi et al. (2023) have explored full-body gesture generation conditioned on speech audio or text transcripts. However, these approaches largely treat raw linguistic input as sufficient semantics, overlooking the deeper communicative intentions—such as emphasis, deixis, and affirmation—that implicitly govern gesture behavior. Recovering these latent functions is essential for generating semantically coherent non-verbal expressions.

Early systems Marsella et al. (2013); Saund & Marsella (2021); McNeill (1992) attempted to derive these deeper communicative functions through rule-based templates, but such approaches lack scalability. Inspired by the linguistic–gesture link in psycholinguistics, we reframe gesture modeling as a reasoning task: the model first understands the reasons or communicative intentions behind the speech, and then treats gestures as their downstream realizations.

To achieve this, we first augment the BEAT-2 dataset Liu et al. (2022b) with structured annotations of inferred communicative functions via VLMs (Vision-Language-Models), creating a large-scale **InG** (Intention-Grounded) dataset. This enriched corpus links speech, inferred intentions, and corresponding gesture realizations, enabling intention-aware gesture generation for the first time.

Leveraging the obtained intention annotations, we propose a CLIP-like gesture understanding model. It learns joint representations of speech rhythm, inferred intentions, and motion dynamics via a hierarchical contrastive alignment framework, enabling it to discern both low-level rhythmic synchronization and high-level communicative goals.

Building upon this understanding model, we introduce the **Intentional Gesture Tokenizer**, a novel quantization model that directly embeds intention semantics into the motion representation space. Unlike prior works that discretize body parts independently without semantic grounding Mughal et al. (2025); Liu et al. (2025a), our tokenizer processes global body motion holistically and supervises latent representations using intention-aware motion features. This design ensures that discrete motion tokens encode not only fine-grained motion patterns but also communicative meaning. With the proposed tokenizer, our approach generates gestures that are not only temporally synchronized but also semantically expressive and interpretable, advancing human-avatar communication. Our contributions can be summarized as follows:

- We formulate gesture generation as an intention-grounded reasoning task, create the InG (Intention-Grounded) dataset by leveraging VLMs to infer communicative functions from speech and augmenting BEAT-2 with structured annotations.

- We introduce the Intentional Gesture Tokenizer, which discretizes global body motion while embedding intention semantics into the latent space through semantic supervision.

- We demonstrate that our method produces gestures that are not only temporally aligned with speech but also semantically meaningful, achieving improved realism and interpretability in human-computer interaction.

## 2  RELATED WORKS

**Co-speech Gesture Generation.** Existing works on co-speech gesture generation typically use skeleton- or joint-level pose representations. Several methods Liu et al. (2022c); Deichler et al. (2023); Xu et al. (2023); Liu et al. (2024a); Zhang et al. (2024a); Liu et al. (2025b) learn hierarchical semantics or apply contrastive learning to align audio and gesture embeddings. HA2G Liu et al. (2022c) constructs multi-level audio-motion embeddings, while TalkShow Yi et al. (2023), CaMN Liu et al. (2022b), and EMAGE Liu et al. (2023) introduce large-scale datasets for joint face-body modeling with GPT-style decoding. More recent models such as MambaTalk Xu et al. (2024), DiffSHEG Chen et al. (2024b), and GestureLSM Liu et al. (2025a) focus on efficient flow matching Li et al. (2025); Zhu et al. (2024) and spatiotemporal modeling. Semantic Gesticulator Zhang et al. (2024b) and RAG-Gesture Mughal et al. (2025) leverage LLMs to retrieve discourse-relevant gestures as references. In contrast, we explore the modeling of communicative intentions as explicit semantics to control gesture generation.

**Vision Tokenization for Generation.** In visual generation Lu et al. (2025); Gao et al. (2024); Lu et al. (2024a), tokenization encodes raw pixels into compact representations Van Den Oord et al. (2017); Rombach et al. (2022). Vector-quantization Van Den Oord et al. (2017) enable discrete latent spaces compatible with autoregressive generation Tian et al. (2024); Sun et al. (2024); Chang et al. (2022); Yu et al. (2023). Recent studies like REPA Yu et al. (2024) and Re.vs.Gen Yao et al. (2025) show that aligning generation representation with understanding improves synthesis quality. We further introduce the **Intentional Gesture Tokenizer**, supervising motion tokenization with intention-aligned semantics, and demonstrate this strategy significantly enhances gesture generation quality.

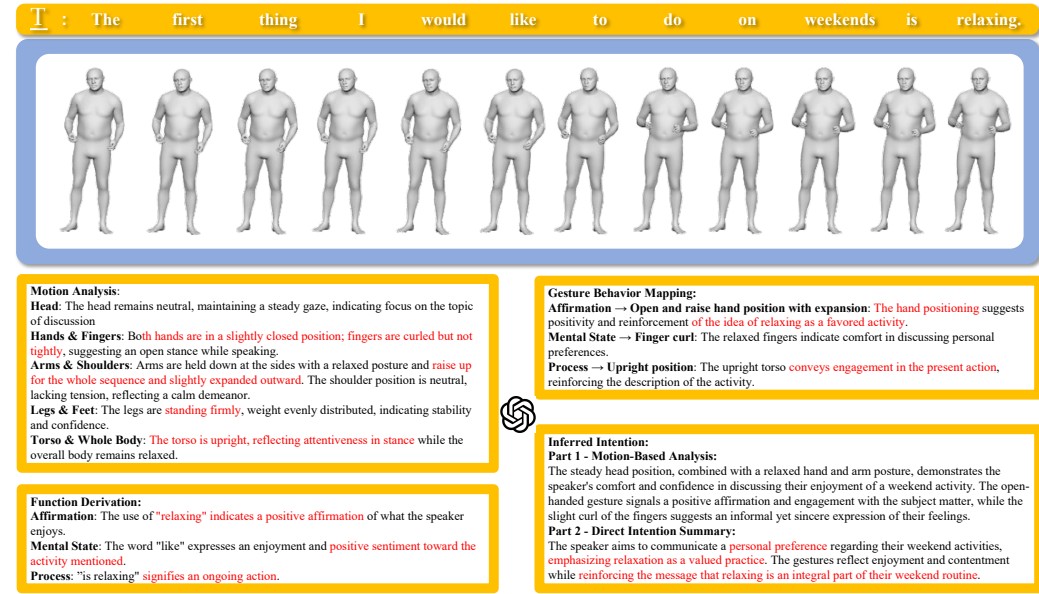

Figure 2: Overview of the annotation pipeline. Sentence-level segments and word-aligned keyframes together with rule-based descriptions are provided to the VLM, which generates (1) motion descriptions across body regions, (2) communicative function labels, (3) gesture behavior mappings, and (4) inferred intentions. This structured annotation enables scalable creation of a semantically grounded, visually aligned gesture dataset.

# 3 INTENTIONAL GESTURE DATASET (ING)

Gestures systematically reflect semantic, pragmatic, and rhetorical structures Marsella et al. (2013); McNeill (1992); Saund & Marsella (2021); for example, affirmation to nodding and comparison to lateral gestures aligned with spatial metaphors. Modeling these communicative functions is critical for generating expressive gestures. Marsella et al. (2013) introduced a rule-based system that derives communicative functions from syntactic and lexical patterns, mapping them to gesture classes via hand-crafted rules. While effective, such systems rely on fixed rules and limited dictionaries, restricting scalability and domain coverage. Building on these insights, we propose a scalable pipeline using VLMs to automatically infer communicicative functions and map them to gestures, enabling large-scale, data-driven semantic gesture modeling. Importantly, we treat these functions as *operational proxies* for communicative intention rather than attempting to recover a single "true" mental state; they are designed as stable, controllable signals for gesture generation.

## 3.1 DATASET CONSTRUCTION

We introduce **InG** (Intention-Grounded Gestures), a dataset that augments BEAT-2 Liu et al. (2022b) and Audio2PhotoReal Ng et al. (2024) with structured, intention-aware annotations linking linguistic functions to gesture behaviors. Unlike prior datasets focused on motion fidelity or prosody alignment, InG enables intention-controllable gesture generation. We employ a two-stage VLM-based strategy, illustrated in Fig. 2.

**Annotation Protocol.** We design a unified annotation pipeline that integrates interpretable rule-based motion descriptors with structured VLM prompting to yield intention-aware gesture annotations. Rather than directly "asking GPT for a label", the pipeline first converts raw motion into constrained, symbolic descriptors, and then uses the VLM only within this structured schema. For training data, the VLM receives sentence context together with word-aligned keyframes and motion descriptors, enabling inference of communicative functions, motion descriptions, and speaker intent. For testing data, annotations are generated solely from transcripts to prevent leakage from observed gestures. Across all segments, we explicitly allow *multi-function* labels (e.g., Emphasis + Deixis + Mental State), reflecting the fact that gesture units often serve multiple discourse functions. Detailed prompting strategies, category definitions, and validation procedures are provided in the Appendix.

**Rule-Based Motion Description Generation.** To provide interpretable motion descriptors aligned with linguistic units, we design a rule-based analysis procedure that converts raw pose sequences into structured gesture descriptions. The process operates in three stages: (1) *Temporal Windowing.* Each sentence is divided into sub-windows of 1–2 seconds based on word-level timestamps. Consecutive words within the same span are grouped together, while longer segments are recursively partitioned. This ensures that each window corresponds to a manageable co-speech gesture unit. (2) *Canonicalized Motion Representation.* All sequences are normalized to the canonical space relative to the body root. For elbows, wrists, head, and fingers, motions are represented by full 3D joint pose angle trajectories or positional change over the window, while for hands, we track 3D positional displacements over time. (3) *Movement Characterization.* Within each window, trajectories are smoothed and segmented by direction and amplitude. Segments are categorized into qualitative tiers (*very slight*, *slight*, *moderate*, *significant*), where thresholds are defined relative to the standard deviation of motion distributions and refined through pilot annotation experiments and human evaluation. Detected patterns are then labeled as monotonic (e.g., "slightly forward"), bidirectional (e.g., "forward then backward"), or oscillatory (e.g., "repeated side-to-side"). Each description is anchored to visual evidence by presenting the first, last, and a representative keyframe from the corresponding word span, ensuring that textual descriptors remain verifiable against pose snapshots. We defer algorithm details in the Appendix.

**Prompt-Based Gesture Intention Annotation.** The rule-based descriptors and reference frames form the grounding for higher-level annotation with GPT-4o-mini. Given segmented utterances and motion summaries, we design a structured prompt for multi-stage annotation: (1) *Motion Analysis* — describing body poses across regions (head, arms, fingers, torso) using the rule-based descriptors together with reference frames; (2) *Function Derivation* — identifying one or more communicative functions (e.g., emphasis, negation, deixis), explicitly allowing multi-label combinations; (3) *Gesture Behavior Mapping* — linking communicative functions to gesture forms; and (4) *Inferred Intention* — synthesizing how verbal and nonverbal cues reflect speaker goals at the level of discourse pragmatics. The model is required to output in a fixed motion–function–intention schema, which constrains free-form generation and reduces hallucinations. This layered approach ensures that gesture annotations are both semantically interpretable and physically grounded, enabling intention-controllable gesture generation.

**Human-in-the-Loop Validation.** To ensure reliability of the VLM-generated annotations, we adopt a human-in-the-loop filtering stage. For each utterance, the LLM is instructed to produce five independent candidate annotations following the structured protocol described above. These candidates vary in phrasing and granularity, but all conform to the same motion–function–intention schema. Human labelers then evaluate the set of five responses and remove those that contain incorrect interpretations, inconsistent mappings, or implausible intentions. Only the remaining high-quality annotations are retained in the dataset. In practice, this combination of rule-based preprocessing, schema-constrained prompting, and human filtering substantially mitigates typical VLM noise and bias while preserving scalability.

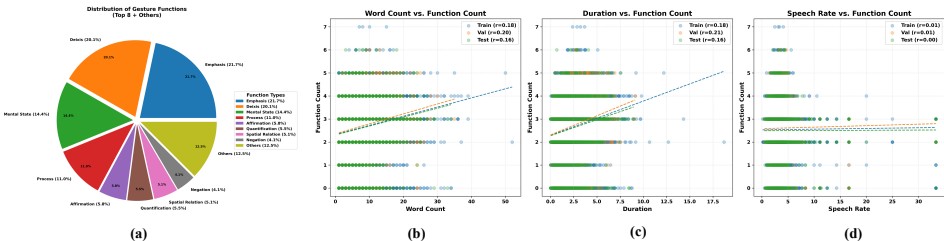

Figure 3: (a) Distribution of communicative function types in InG dataset. (b-d) Correlations between function count and utterance-level features: word count, duration, and speech rate. Function count correlates positively with utterance length and duration, but minimally with speech rate.

## 3.2 DATASET STATISTICS AND ANALYSIS

Our InG dataset includes 34,641, 3,598, and 9,674 annotated utterances for train/val/test splits, respectively, each enriched with motion-grounded or intent-inferred communicative functions. We defer the specific statistics of the distribution of 16 function types, covering pragmatic categories such as Emphasis, Deixis, Mental State, and Process in the Appendix.

As shown in Fig. 3 (a), Emphasis (21.7%) and Deixis (20.1%) are the most prevalent functions, followed by Mental State (14.4%) and Process (11.0%). We further analyze correlations between linguistic features and function density (Fig. 3, b-d). Function counts correlate positively with utterance length (r = 0.18–0.20) and duration (r = 0.16–0.21), but show minimal correlation with speech rate (r ≈ 0.00–0.01). This suggests that communicative function density scales with the informational content rather than speaking speed, aligning with psycholinguistic findings.

While function derivations follow a well-defined ontology, gesture behavior mappings and inferred intentions exhibit open-ended variability across speakers and contexts. To ensure interpretability, our prompt design grounds mappings in established literature (McNeill McNeill (1992), Kendon Saund & Marsella (2021)). Additional statistics and analysis are provided in Appendix.

**Annotation Validation Study.** We evaluate annotation quality through a pairwise human preference study on 100 randomly sampled utterances. Each utterance is annotated using: (1) our train-style VLM protocol (motion input), (2) our test-style VLM protocol (transcript only), and (3) a free-form human annotation. For each utterance, two comparisons are created: VLM (train-style) vs. human, and VLM (test-style) vs. human.

The human baseline is produced by two expert annotators (non-authors) with prior exposure to gesture and multimodal communication literature. For each clip, they are given the speech transcript, rendered motion, and the same communicative function ontology as our model, and asked to write free-form descriptions of gesture functions and intentions without seeing any VLM outputs.

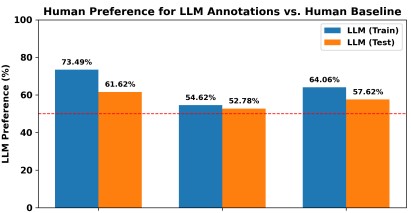

Figure 4: Pairwise preference for VLM vs. human annotations across three evaluation settings. Blue: train-style prompting (with motion); Orange: test-style (transcript only). Red dashed line indicates chance (50%).

Three raters independently judge each comparison across three criteria: **E1: Intent Alignment** (which better captures communicative intent), **E2: Gesture Relevance** (which provides more plausible gestures), and **E3: Overall Preference** (clarity and function-gesture alignment).

Final preferences are determined by majority vote. Shown in Fig. 4, both VLM protocols outperform the free-form human baseline. Inter-rater agreement averaged 0.76 Fleiss' $\kappa$, indicating substantial consistency. This comparison shows that our structured motion–function–intention pipeline yields annotations that are more consistent and directly usable for modeling than free-form text.

## 4 INTENTIONAL GESTURE GENERATION

### 4.1 INTENTION UNDERSTANDING

To leverage textual intention for co-speech gesture synthesis, we propose a CLIP-like understanding model, H-AuMoCLIP, for gestures to better encode textual intention features. We build this Intention-Audio-Motion CLIP on top of TANGO's (Liu et al., 2024a) audio-motion alignment model (AuMoCLIP) by explicitly incorporating communicative intent.

**Intentional Semantic Fusion.** AuMoCLIP trains its joint embedding space using contrastive learning, where its audio tower combines acoustic features with BERT-encoded speech transcripts. Building upon this, we introduce Intentional Semantic Fusion for merging intention semantics to create a richer, intention-aware joint embedding space. Our Intentional Semantic Fusion mechanism fuse transcript and intention embeddings via a linguistically grounded mechanism. Following TANGO, we align transcript tokens to the audio timeline using a CTC-based model, and encode both the transcript and annotated intent with separate BERT encoders. In addition to TANGO, the aligned transcript embeddings serve as queries in a cross-attention module, with intention embeddings as keys and values. The output captures context-specific communicative cues, which are then concatenated with wav2vec2 audio features to form the final high-level representation. This composite embedding is used in contrastive training to jointly align audio, motion, and intent.

**Functionality.** As in Fig.5, the resulting encoders improve generation by serving as two roles: (1) the motion encoder provides semantic supervision for gesture tokenization (Sec. 4.2), and (2) the audio encoder conditions gesture generation with both rhythmic and intentional signals (Sec. 4.3).

Figure 5: **Overview of our intentional gesture generation framework.** Left: H-AuMoCLIP learns a hierarchical joint embedding of motion, audio, and intention. Transcript embeddings (BERT) aligned via CTC serve as queries in a cross-attention module with intention embeddings as keys/values. The resulting semantic features are concatenated with wav2vec2 audio features for contrastive learning. Right: Motion is quantized via a multi-codebook VQ module and supervised by semantic features from H-AuMoCLIP, enabling expressive and controllable gesture generation.

## 4.2 INTENTIONAL GESTURE TOKENIZER

Prior works Mughal et al. (2025); Liu et al. (2025a; 2023) model different body regions using separate encoders, decoders, and codebooks per body part. While this design encourages local disentanglement, it has three limitations: (1) high training and inference cost, (2) poor global body coherence, and (3) weak connection to speech-level semantics. To address these issues, we propose an **Intentional Gesture Tokenizer** with two core innovations: (1) learning disentangled latent factors within a unified global body representation through multi-codebook quantization, and (2) embedding explicit intention semantics into the motion representation through semantic supervision.

**Latent Multi-Codebook Quantization.** Instead of separate body-specific quantizers, we discretize the global motion latent $f \in \mathbb{R}^d$ using a set of $n$ independent codebooks. The latent is partitioned into $n$ chunks $\{f_1, f_2, \ldots, f_n\}$, each quantized separately:

$$\hat{f} = \text{Concat}\left(\mathcal{Q}(Z_1, f_1), \ldots, \mathcal{Q}(Z_n, f_n)\right), \tag{1}$$

where $\mathcal{Q}$ denotes vector quantization. This design maintains global body context while allowing structured disentanglement to emerge across codebooks during training. Unlike prior methods, our tokenizer processes the full body motion jointly, enabling better modeling of coordination patterns necessary for communicative gestures.

**Intentional Semantic Supervision.** To embed contextual intent into motion representations, we supervise the quantized latents using features from the pretrained H-AuMoCLIP motion encoder (Sec. 4.1). A linear projection is applied to the quantized output $Z$ to match the dimension of reference features $F$, and we compute a temporal cosine margin loss:

$$\mathcal{L}_{\text{sem}} = \frac{1}{T} \sum_{t=1}^{T} \text{ReLU}\left(1 - \frac{z_t' \cdot f_t}{\|z_t'\|\|f_t\|}\right), \tag{2}$$

where $z_t'$ and $f_t$ are projected quantized latents and motion encoder features at timestep $t$. This supervision enforces semantic alignment with high-level intention cues derived from speech.

**Training Objective.** Our full training objective includes: (i) a reconstruction loss $\mathcal{L}_{\text{R}}$, (ii) a vector quantization loss $\mathcal{L}_{\text{VQ}}$, and (iii) the proposed semantic supervision loss $\mathcal{L}_{\text{sem}}$:

$$\mathcal{L} = \mathcal{L}_{\text{R}} + \lambda_{\text{VQ}}\mathcal{L}_{\text{VQ}} + \lambda_{\text{sem}}\mathcal{L}_{\text{sem}}. \tag{3}$$

We empirically set $\lambda_{\text{sem}} = 1$ and $\lambda_{\text{VQ}} = 0.25$. This balanced formulation ensures that quantized motion tokens preserve both fine-grained fidelity and semantic coherence across time.

## 4.3 GESTURE GENERATION

We then leverage Sec.4.1 and Sec.4.2 to enhance gesture generation. We adapt GestureLSM (Liu et al., 2025a), by replacing its original audio encoder and motion tokenizer. Specifically, we replace GestureLSM's audio encoder with our intent-aware audio encoder, which extracts hierarchical representations capturing both rhythmic audio features and communicative intentions. Furthermore,

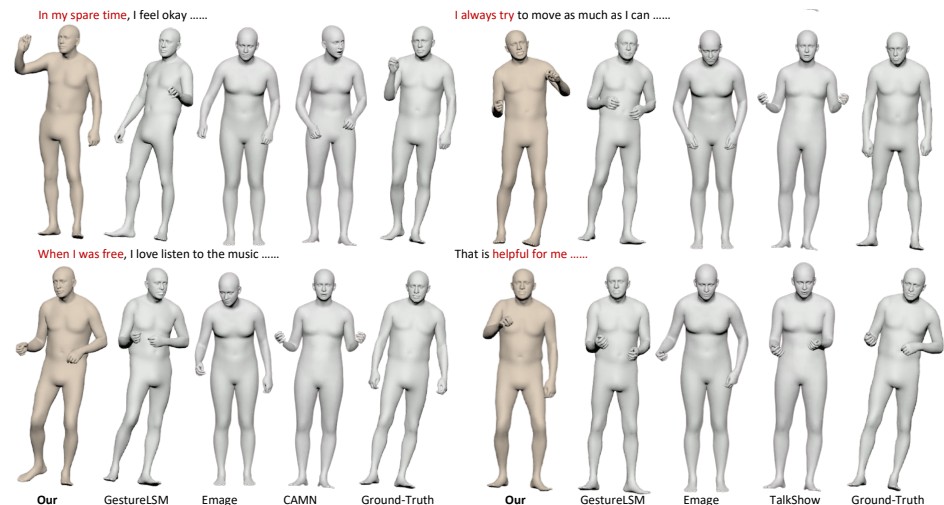

Figure 6: **The subjective Comparisons.** Compared with others, Intentional-Gesture presents more natural and coherent motion patterns to represent specific words or phrases (highlighted in red).

Table 1: **Comparison with state-of-the-art methods trained on BEAT-2.** We demonstrate superior performance, especially when generalizing across multiple speaker identities.

| | 1 Speaker | | | All Speakers | | | Novel 5 speakers | | |
|---|---|---|---|---|---|---|---|---|---|
| | FGD↓ | BC→ | Div.→ | FGD↓ | BC→ | Div.→ | FGD↓ | BC→ | Div.→ |
| GT | | 0.703 | 11.97 | | 0.477 | 7.29 | | 0.671 | 8.93 |
| CaMN Liu et al. (2022b) | 0.604 | 0.676 | 9.97 | 0.512 | 0.200 | 5.58 | 0.812 | 0.563 | 6.71 |
| EMAGE Liu et al. (2023) | 0.570 | 0.793 | 11.41 | 0.692 | 0.284 | 6.06 | 0.936 | 0.643 | 7.47 |
| Audio2Photoreal Ng et al. (2024) | 1.02 | 0.550 | **12.47** | 0.849 | 0.326 | 6.24 | 1.012 | 0.464 | 5.97 |
| RAG-Gesture Mughal et al. (2025) | 0.879 | 0.730 | 12.62 | 0.447 | **0.471** | 9.03 | - | - | - |
| GestureLSM Liu et al. (2025a) | 0.408 | 0.714 | 13.24 | 0.446 | 0.525 | 9.23 | 0.664 | 0.621 | 10.45 |
| Ours | **0.379** | **0.690** | 11.00 | **0.256** | 0.534 | **6.68** | **0.441** | **0.686** | **9.39** |

we replace its RVQ-based tokenizer with our Intentional Gesture Tokenizer. Additional architectural details of the original GestureLSM are deferred to the Appendix.

## 5 EXPERIMENTS

We conduct main experiments on BEAT2 Liu et al. (2023), which comprises 60 hours of high-quality SMPL-based conversational or speach gesture data collected from 25 speakers. The dataset contains 1,762 sequences, each with an average duration of 65.66 seconds, following the train-validation-test split protocol defined in EMAGE Liu et al. (2023). In addition, we also explores its application on Audio2Photoreal Ng et al. (2024), which provides about 8 hours of dyadic interactions between listening and speaking actions. We defer the experiment results on Audio2PhotoReal to Appendix with further video demos provided in the supplementary material.

### 5.1 QUANTITATIVE COMPARISONS

**Metrics.** We evaluate Fréchet Gesture Distance (FGD) Yoon et al. (2020) for pose sequence angle distributional similarity, Diversity (Div.) Li et al. (2021a) as the average L1 distance across clips, and Beat Constancy (BC) Li et al. (2021b) for speech-motion synchronization.

**Evaluation Results.** We evaluate both quality and generalization in three settings: single-speaker, all-speaker, and zero-shot (training on 20 speakers and testing on 5 unseen speakers). As shown in Tab. 1, our method significantly outperforms baselines across all settings. These gains are attributed to our intentional motion representation and intention control, which reduce unnatural gesture patterns. More detailed comparisons are provided in the Appendix. To validate runtime efficiency, we generate intention descriptions from transcripts via an API prior to model execution. Our method retains the efficiency of GestureLSM Liu et al. (2025a), while improving performance through a unified motion representation, and achieves generation at 30.4 FPS on an H100 GPU.

## 5.2 QUALITATIVE COMPARISONS

**Evaluation Results** As depicted in Fig. 6, our approach generates gestures that exhibit improved rhythmic alignment and a more natural appearance compared to existing methods. For example, when conveying *"helpful for me"*, our method directs the subject to extend left hand forward, effectively representing the intention of an explanation. In contrast, competing methods fail to capture this nuance, often generating static or unnatural poses where one or both arms remain down.

**User Study.** We conducted a user study with 20 participants and 160 video samples, 40 from each of GestureLSM Liu et al. (2025a), EMAGE Liu et al. (2023), CAMN Liu et al. (2022b), and Ours. Each participant viewed the videos in a randomized order and rated them on a scale of 1 (lowest) to 5 (highest) based on three criteria: (1) *realness*, (2) speech-gesture *synchrony*, and (3) *smoothness*. As shown in Tab. 2, Our method outperforms other methods across all criteria, achieving higher Mean Opinion Scores (MOS). We follow a standard 1–5 Likert MOS scale with textual anchors (1 = very poor, 3 = acceptable, 5 = excellent), and compute MOS as the average rating across raters and clips for each method. Detailed statistics (mean $\pm$ std, 95% confidence intervals, and significance tests) are reported in Appendix. F.

Table 2: Subjective evaluation shown as Mean Opinion Scores (MOS).

| Methods | $MOS_1$ | $MOS_2$ | $MOS_3$ |
|---|---|---|---|
| EMAGE | 2.01 | 2.42 | 2.31 |
| CAMN | 1.34 | 2.23 | 2.14 |
| GestureLSM | 3.43 | 3.61 | 3.48 |
| Ours | **3.76** | **4.11** | **3.92** |

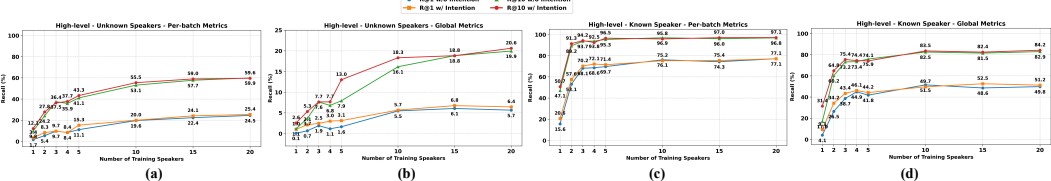

Figure 7: **Hierarchical Alignment Analysis.** Retrieval-based evaluation of H-AuMoCLIP across semantic features. We report Recall@1/10 under per-batch and global settings for both known and unknown speakers. Incorporating intentional semantics improves retrieval performance with few training speakers. Increasing speaker diversity further enhances generalization for retrieval.

## 6 ABLATION STUDIES AND ANALYSIS

We explore the hierarchical audio-intention and motion alignment, the design of tokenizer and generator, the analysis of the quality of annotation effects on learning, in this section. We defer sequence length effect, quantizer analysis to Appendix.

### 6.1 EVALUATION OF HIERARCHICAL AUMOCLIP

We evaluate H-AuMoCLIP using **retrieval-based metrics** that measure how well audio and intentional features align with corresponding motion features. We report Recall@1 and 10 under two settings: **per-batch** (within batch of 128) and **global** (across the entire test set). The retrieval evaluates sequence-level alignment using mean-pooled audio and motion features for full-clip matching. We evaluate generalization to both **seen** (in-domain) and **unseen** (out-of-domain) speakers, assessing robustness across speaker identities.

**Intention-Aware Alignment Improves Performance.** Incorporating intentional semantics consistently improves retrieval, especially under low-data regimes (1–5 speakers) (Fig. 7). High-level alignment shows larger gains compared to low-level retrieval, confirming that explicit intention modeling better captures sequence semantics.

**Speaker Diversity Enhances Generalization.** Training on multiple speakers significantly improves generalization to unseen speakers. Even with the same total training data, models trained on more speaker identities achieve better high-level retrieval, suggesting that speaker diversity encourages learning more robust gesture-speech mappings, as shown in Appendix.

Table 3: **Quantitative Ablation Results.** (Left) Quantizer ablation comparing design variants and levels of intentional supervision. (Right) Gesture generation improvements with architectural modules, semantic controls and sensitivity analysis of the whole generator model.

| Model | rFGD↓ | Utility↑ | L1↓ | FGD↓ | BC→ | Div.→ |
|---|---|---|---|---|---|---|
| EMAGE (VQ) | 0.0867 | 34.35 | 0.3643 | 0.512 | 0.623 | 7.76 |
| RAG-Gesture (Contin.) | 0.0423 | - | 0.448 | 0.423 | 0.625 | 8.64 |
| ProbTalk (PQ) | 0.0283 | 53.24 | 0.3283 | 0.451 | 0.656 | 7.96 |
| GestureLSM (RVQ) | 0.0012 | 27.56 | **0.1557** | 0.314 | 0.527 | 8.64 |
| *Ours (Multi-VQ)* | | | | | | |
| only transformer | 0.0021 | 97.25 | 0.3043 | 0.332 | 0.474 | 8.01 |
| only CNN | 0.0013 | 98.43 | 0.2656 | 0.348 | 0.671 | 7.96 |
| Optimal Quantizer Design | **0.0011** | **98.84** | 0.2468 | 0.315 | 0.512 | 7.67 |
| +Low-level Supervision | 0.0014 | 96.72 | 0.2566 | 0.295 | **0.477** | 5.96 |
| +High-level (seq) | 0.0014 | 96.68 | 0.2512 | 0.286 | 0.614 | 6.21 |
| +High-level (temporal) | 0.0012 | 97.17 | 0.2488 | **0.256** | 0.534 | **6.68** |

| Model | FGD↓ | BC→ | Div.→ |
|---|---|---|---|
| GestureLSM Liu et al. (2025a) | 0.446 | 0.525 | 9.23 |
| + DiT Peebles & Xie (2023) | 0.354 | 0.523 | 6.63 |
| + Intention Condition | 0.339 | 0.545 | 6.44 |
| + H-AuMoCLIP | 0.314 | 0.527 | 8.64 |
| + Intentional-Tokenizer | **0.256** | **0.534** | **6.68** |
| *Sensitivity Analysis* | | | |
| Re-summarized intent | 0.256 | 0.533 | 6.66 |
| Unmatched intent | 0.284 | 0.521 | 6.47 |
| Window = 0.7 | 0.257 | 0.533 | 6.64 |
| Window = 0.5 | 0.256 | 0.533 | 6.67 |
| w/o intent text | 0.281 | 0.523 | 6.54 |

Table 4: **Ablation on Dataset Annotation.** (Left) Structured intention annotation improves both retrieval and generation, with training-time annotation yielding the best overall results. (Right) Intention provides the strongest generation quality, motion description enhances retrieval, and combining all signals achieves the best overall recall but a trade-off in generation.

| Setting | Retrieval | | | Generation | | |
|---|---|---|---|---|---|---|
| | R@1↑ | R@5↑ | R@10↑ | FGD↓ | BC→ | Div.→ |
| N/A | 8.37 | 25.41 | 35.86 | 0.284 | 0.543 | 6.74 |
| Baseline | 8.18 | 23.41 | 33.59 | 0.269 | 0.612 | **6.79** |
| Train-Set | **8.47** | **28.43** | **37.67** | **0.256** | **0.534** | 6.68 |
| Test-Set | 8.44 | 28.21 | 37.64 | 0.262 | 0.545 | 6.56 |

| Setting | Retrieval | | | Generation | | |
|---|---|---|---|---|---|---|
| | R@1↑ | R@5↑ | R@10↑ | FGD↓ | BC→ | Div.→ |
| A | 8.37 | 25.41 | 35.86 | 0.284 | 0.543 | 6.74 |
| A + M | 9.21 | 29.76 | 38.43 | 0.464 | 0.412 | 4.78 |
| A + I | 8.47 | 28.43 | 37.67 | **0.256** | **0.534** | **6.68** |
| A + I + M | **9.41** | **31.62** | **40.59** | 0.343 | 0.565 | 7.44 |

## 6.2 EVALUATION OF INTENTIONAL GESTURE TOKENIZER

We evaluate our intentional gesture tokenizer using **reconstruct FGD (rFGD)**, **Codebook Utility**, and **L1 error**, with generation metrics reported in Sec. 5.1. We defer further analysis in Appendix.

**Tokenizer Comparison.** We leverage the same generator design in our framework with different latent representation from various tokenizer designs for this comparison. As in Tab. 3 Left, our Intentional Gesture Tokenizer achieves a significantly lower rFGD and higher diversity compared to previous tokenizers Liu et al. (2023; 2024b); Mughal et al. (2025); Liu et al. (2025a).

**Effect of Intentional Semantic Supervision.** We design three supervision strategies using H-AuMoCLIP features: (a) CNN-layer supervision for low-level spatial detail, (b) transformer-layer with mean pooling for high-level sequence supervision, and (c) transformer-layer with frame-level temporal supervision. Shown in Tab. 3 (Left), integrating intentional supervision into motion token learning significantly improves generation quality, while maintaining strong reconstruction fidelity. Notably, frame-level temporal supervision achieves the best, demonstrating that temporal alignment of intention and motion is key to learning expressive gesture representations.

## 6.3 EVALUATION OF GESTURE MOTION GENERATION

**Design Analysis.** Replacing GestureLSM with DiT Peebles & Xie (2023) based architecture improves the generation performance. We further investigate how intention representations impact gesture generation. As a baseline, we condition the generator on sentence-level BERT embeddings, which provides only marginal improvement (Tab. 3, Right). Replacing these with intention embeddings derived from H-AuMoCLIP yields substantial gains in FGD and diversity.

**Sensitivity Analysis.** We evaluate the robustness of our model under noisy or mismatched intention inputs. Using LLM-rewritten paraphrases of intention descriptions yields negligible differences in FGD and diversity, suggesting that the model is not sensitive to surface-level textual variation. When using temporally unmatched or missing intention text, FGD degrades slightly (by 0.025–0.03), and synchronization weakens marginally, but the model remains stable overall.

## 6.4 EVALUATION OF DATASET ANNOTATION

**Impact of Intention Annotation on Alignment and Generation.** We assess the impact of intention annotation quality via an ablation study (Tab. 4, left), comparing four settings: (1) no annotation, (2) baseline unstructured annotation, (3) our structured pipeline during training, and (4) structured annotation also applied at test time (transcript only). We discover any annotation improves

Table 5: Semantic case study on BEAT2. We stratify segments into neutral/beat-like, iconic, and metaphoric gestures and compare a baseline audio+text model with our intention-conditioned model in terms of intention-aware retrieval (R@k) and generation metrics (FGD, BC, Div.).

| Semantic type | Method | R@1 ↑ | R@5 ↑ | R@10 ↑ | FGD ↓ | BC → | Div. → |
|---|---|---|---|---|---|---|---|
| Neutral / beat-like | Baseline | 8.79 | 28.47 | 40.52 | 0.144 | 0.523 | 4.17 |
| | +Intention | **10.23** | **34.22** | **41.37** | **0.132** | **0.565** | **7.29** |
| Iconic | Baseline | **7.44** | 24.97 | 38.75 | 0.223 | **0.482** | **4.61** |
| | +Intention | 7.23 | **25.12** | **39.01** | **0.212** | 0.471 | 5.23 |
| Metaphoric | Baseline | 7.21 | 20.20 | 38.57 | 0.201 | **0.512** | 6.14 |
| | +Intention | **8.01** | **23.22** | **40.44** | **0.197** | 0.488 | **5.44** |

generation, though baseline signals slightly harm retrieval, suggesting noisy semantics still guide alignment. Our structured training annotations achieve the best overall results, and adding test-time annotation further boosts diversity and balance. The moderate gap between baseline and structured settings indicates robustness to noise, while highlighting clear gains from high-quality supervision.

**Comparison of Intention And Motion Description as Control.** We compare audio (A), motion description (M), and intention (I) as input signals (Tab. 4, right). Adding motion description (A+M) improves retrieval but reduces generation diversity, while intention (A+I) gives the best overall generation. Combining all three (A+I+M) achieves the strongest retrieval, though generation is slightly less balanced than A+I alone. This shows motion description mainly strengthens alignment, while intention provides higher-level semantic control for natural and diverse generation.

## 6.5 SEMANTIC CONSISTENCY EVALUATION

Standard objective metrics such as FGD, Diversity, and Beat Constancy primarily capture motion realism and rhythm, but not directly communicative effectiveness. We therefore report semantic consistency metrics that operate in the intention-aware representation space and complement the main quantitative results.

**Intention-aware audio–motion retrieval.** We use H-AuMoCLIP to embed (audio, transcript, intention) triplets and motion sequences into a shared space. Given a query composed of audio + transcript + intention, the task is to retrieve the corresponding motion segment from a candidate pool. We report Recall@k (R@1, R@5, R@10) on the test set. Models trained with intention supervision consistently obtain higher R@k than audio/text-only variants, indicating that the learned representations capture function-level semantics rather than only kinematic similarity.

**Case study: iconic vs. metaphoric vs. neutral segments.** To further probe semantic consistency, we conduct a small case study by stratifying segments according to their gesture semantics. The comparison is based on the semantic label of short segments similar to the setting in short sequence generation from Tab.15 from BEAT2: (i) *Neutral / beat-like*, (ii) *Iconic*, and (iii) *Metaphoric* For each type, we compare a baseline model conditioned on audio + transcript with our intention-conditioned model (audio + transcript + intention) in terms of both retrieval and generation quality. We observe consistent gains from intention supervision across all three types, with especially clear relative improvements on iconic and metaphoric segments where gesture meaning is tightly coupled to speech. Metaphoric and iconic present slightly worse performance due to variability of the data.

## 7 CONCLUSION

We present **Intentional Gesture**, a novel framework that formulates gesture generation as an intention reasoning task grounded in high-level communicative functions. We first curate the **InG** dataset with structured annotations of latent intentions—such as emphasis, comparison, and affirmation—extracted via large language models. Then, we introduce the **Intentional Gesture Tokenizer**, which discretizes global motion while embedding intention semantics into the representation space through targeted supervision. By bridging speech semantics and motion behaviors, our method produces gestures that are both temporally synchronized and semantically meaningful. Experiments demonstrate strong improvements in gesture quality, offering a controllable and modular foundation for expressive gesture generation in digital humans and embodied AI.

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

# Intentional Gesture: Deliver your Intentions with Gestures for Speech

## Supplementary Material

## A OVERVIEW

The supplementary material is organized into the following sections:

- Section B: Additional Dataset Analysis
- Section C: Annotation Protocol and Validation
- Section D: Implementation Details
- Section E: Additional Experiments
- Section F: User Study Details
- Section G: Metric Details
- Section H: Ethical Statement
- Section I: Reproducibility statement
- Section J: The Use of Large Language Models
- Section K: Limitations

For more visualization, please see the additional demo videos.

## B ADDITIONAL DATASET ANALYSIS

### B.1 FUNCTION-TO-GESTURE MAPPING GROUNDING

Our function-to-gesture mappings derive from established frameworks in gesture pragmatics, particularly McNeill McNeill (1992) and Kendon Kendon (2004). Tab. 6 presents gesture forms associated with each communicative function, which inform our VLM annotation prompt's gesture behavior mapping.

Certain functions correspond to consistent physical gestures (e.g., Deixis to pointing, Emphasis to beat gestures, Negation to head shakes), while others like Modal or Mental State manifest in subtler movements (fist tightening, shoulder shrugs). These literature-backed correspondences ensure interpretable and plausible annotations, providing a bridge between gesture generation and discourse semantics.

Tab. 6 shows the function distribution across dataset splits. Core functions such as Deixis (57-61%), Emphasis (46-51%), Mental State ( 41%), and Process (26-29%) are well-represented with minimal variation across splits. Less frequent functions like Comparison, Modal, and Valence (5-8%) and specialized functions (Intensifier, Physical Relation, ¡2%) show distributional consistency. Note that these percentages reflect per-sentence function occurrence rather than the cumulative distribution reported in the main paper.

### B.2 CO-OCCURRENCE PATTERNS AND SPEAKER-SPECIFIC GESTURE PROFILES.

To further examine the structure of our function annotations, we analyze co-occurrence patterns and speaker-level gesture usage. Figures 8(a–c) present conditional co-occurrence heatmaps for the top 8 gesture functions across train, validation, and test splits. Each cell reflects the probability that function $j$ co-occurs given function $i$ within the same utterance. We observe strong mutual co-occurrence between Emphasis and Deixis, as well as between Mental State and Emphasis, suggesting these functions often emerge in jointly expressive speech segments. These co-occurrence trends remain stable across dataset splits, reinforcing the semantic consistency of our annotations.

Figure 8(d) shows a radar plot of gesture function usage for the top 6 most frequent speakers. While some functions like Deixis and Emphasis are commonly expressed across speakers, other functions (e.g., Contrast, Modal, Quantification) exhibit speaker-specific variability. This aligns with prior

Table 6: Gesture function statistics and mappings. For each function, we report its relative frequency (%) across dataset splits and its typical gestural manifestation.

| Function | Frequency (%) | | | Typical Gesture Mapping |
|---|---|---|---|---|
| | Train | Val | Test | |
| Deixis | 57.3 | 61.8 | 60.2 | Index finger pointing, gaze direction shift |
| Emphasis | 48.3 | 50.6 | 46.4 | Beat gestures, small head nods |
| Mental State | 42.0 | 41.1 | 41.1 | Shrug, slow head tilt, hand on chest |
| Process | 29.1 | 25.6 | 28.8 | Circular motion, continuous hand movement |
| Quantification | 16.7 | 20.6 | 17.0 | Spread fingers, repeated motions |
| Spatial Relation | 16.1 | 16.5 | 18.2 | Hands indicating space or depth |
| Negation | 13.2 | 12.3 | 11.0 | Head shake, subtle hand wave |
| Affirmation | 8.9 | 10.7 | 9.9 | Big nod, repeated nods |
| Valence | 8.1 | 7.0 | 7.1 | Open hands (positive), recoiling motion (negative) |
| Modal | 7.6 | 8.1 | 5.2 | Tight fist, upward palm with tension |
| Comparison | 6.6 | 7.7 | 5.6 | Left-right hand sweep, comparative spacing |
| Interrogative | 4.6 | 2.9 | 3.4 | Raised eyebrows, open palms |
| Contrast | 3.9 | 3.5 | 3.2 | Alternating hand gestures, lateral head tilt |
| Intensifier | 1.4 | 1.4 | 1.2 | Sharp eyebrow raise, large gesture amplitude |
| Performance Factor | 1.0 | 1.1 | 0.9 | Gaze aversion, short blink, pause gestures |
| Physical Relation | 0.6 | 0.6 | 0.7 | Gesture showing size/shape (e.g., distance between hands) |

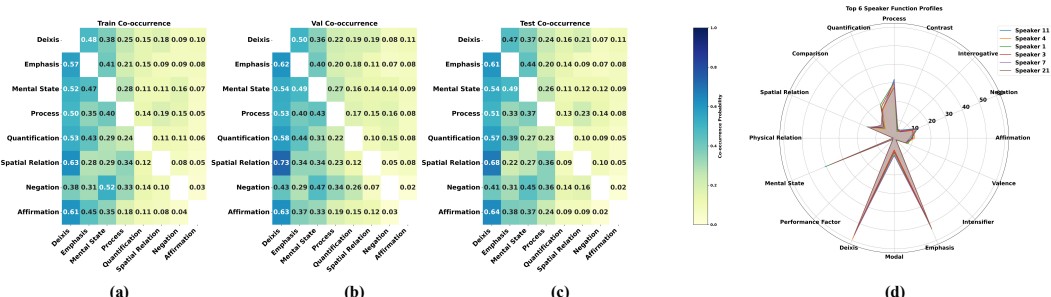

(a)  (b)  (c)  (d)

Figure 8: **Co-occurrence and speaker-level analysis of gesture function annotations.** **(a–c)** show conditional co-occurrence heatmaps of the top 8 gesture functions across the train, validation, and test splits, respectively. Each cell indicates the probability of function $j$ appearing given function $i$ (i.e., $P(j|i)$). Strong pairings (e.g., Emphasis + Deixis, Mental State + Emphasis) reveal compositional gesture semantics. **(d)** presents radar plots of function distribution across the top 6 speakers, revealing shared trends (e.g., high Deixis usage) and speaker-specific variation in gesture behavior.

findings that gesture behavior reflects both discourse demands and speaker idiosyncrasies Kendon (2004). Such variation presents a valuable modeling challenge for systems that aim to personalize or adapt gesture generation to individual styles.

---

**Algorithm 1** Motion Pattern Detection

---

**Require:** Input data $\mathbf{y} \in \mathbb{R}^T$, thresholds $\epsilon_s, \epsilon$
**Ensure:** Motion statistics and extrema relations
1: $\mathbf{y} \leftarrow$ reshape to 1D array
2: **if** $T \leq 1$ **then return** insufficient_data
3: **end if**
4: **// Extract key statistics**
5: $y_0, y_T \leftarrow \mathbf{y}[0], \mathbf{y}[T-1]$
6: $i_{max}, i_{min} \leftarrow \arg\max(\mathbf{y}), \arg\min(\mathbf{y})$
7: $y_{max}, y_{min} \leftarrow \mathbf{y}[i_{max}], \mathbf{y}[i_{min}]$
8: $\delta \leftarrow y_{max} - y_{min}, \Delta \leftarrow y_T - y_0$
9: **// Check if motion is static**
10: **if** $\delta < \epsilon_s$ **then**
11:     **return** {pattern: 'linear', range: $\delta$, direction: $\text{sign}(\Delta)$}
12: **end if**
13: **// Compute extrema relations**
14: $\mathbf{s} \leftarrow [|y_0 - y_{max}| \leq \epsilon, |y_0 - y_{min}| \leq \epsilon]$               $\triangleright$ Start position
15: $\mathbf{e} \leftarrow [|y_T - y_{max}| \leq \epsilon, |y_T - y_{min}| \leq \epsilon]$               $\triangleright$ End position
16: $\mathbf{in} \leftarrow [i_{max} \notin \{0, T-1\}, i_{min} \notin \{0, T-1\}]$        $\triangleright$ Interior extrema
17: **return** $\{\mathbf{y}, \Delta, \delta, \mathbf{s}, \mathbf{e}, \mathbf{in}\}$

---

**Algorithm 2** Motion Pattern Classification

---

**Require:** Extrema relations $\mathbf{s} = [s_{max}, s_{min}], \mathbf{e} = [e_{max}, e_{min}], \mathbf{in} = [in_{max}, in_{min}]$
**Ensure:** Pattern type and description
1: **if** $(s_{max} \wedge e_{min}) \vee (s_{min} \wedge e_{max})$ **then**
2:     pattern $\leftarrow$ 'round_trip'                    $\triangleright$ Opposite extremes
3: **else if** $(s_{max} \vee s_{min}) \wedge (e_{max} \vee e_{min})$ **then**
4:     pattern $\leftarrow$ 'return_to_extreme'            $\triangleright$ Same extreme
5: **else if** $(s_{max} \vee s_{min}) \wedge \neg(e_{max} \vee e_{min})$ **then**
6:     pattern $\leftarrow$ 'peak_at_start'              $\triangleright$ Leave from extreme
7: **else if** $(e_{max} \vee e_{min}) \wedge \neg(s_{max} \vee s_{min})$ **then**
8:     pattern $\leftarrow$ 'peak_at_end'                $\triangleright$ Arrive at extreme
9: **else if** $in_{max} \wedge in_{min}$ **then**
10:     pattern $\leftarrow$ 'peak_between'             $\triangleright$ Both extremes inside
11: **else if** $in_{max} \oplus in_{min}$ **then**
12:     pattern $\leftarrow$ 'single_extreme_inside'       $\triangleright$ One extreme inside
13: **else**
14:     pattern $\leftarrow$ 'complex_extrema'          $\triangleright$ Boundary-aligned
15: **end if**
16: **return** pattern

---

**Algorithm 3** Helper Functions

---

1: **function** GETDIRECTION($\Delta$)
2:     **return** $\begin{cases} \text{'positive'} & \text{if } \Delta > 0 \\ \text{'negative'} & \text{if } \Delta < 0 \\ \text{'none'} & \text{otherwise} \end{cases}$
3: **end function**

4: **function** CLASSIFYMOVEMENT($\Delta, \epsilon_s, \epsilon_{slow}$)
5:     **return** $\begin{cases} \text{'static'} & \text{if } |\Delta| < \epsilon_s \\ \text{'slow'} & \text{if } |\Delta| < \epsilon_{slow} \\ \text{'significant'} & \text{otherwise} \end{cases}$
6: **end function**

---

Table 7: **Training-time annotation prompt with visual grounding.** Our framework analyzes human gestures by integrating visual keyframes with speech.

---

**System Prompt:**
Assume you are the annotator for human gestures. Given images for each word the person speaks, you need to provide fine-grained analysis from motion captions, to Function Derivations, Gesture Behavior Mapping, and finally Inferred Intention. The Definition of Function Derivation & Gesture Behavior Mapping are as follows:

[Function Derivation: 16 classes of Function Derivations]
[Gesture Behavior Mapping: How functions map to physical movements.]

**User Prompt:**
I will provide you with a transcript of speech, the atomic pose angle movement descriptions and corresponding images showing the speaker's gestures. Please analyze the motion and provide a detailed description as the generation output following this format:

[Format Instruction]
**Motion Analysis:**
- **Head:** Describe head movements (nodding, shaking, tilting)
- **Hands & Fingers:** Describe hand gestures, positions, finger articulations
- **Arms & Shoulders:** Describe arm movements and shoulder positions
- **Legs & Feet:** Describe lower body movements and weight shifts
- **Torso & Whole Body:** Describe posture and body orientation
**Function Derivation:** List relevant functions from the prior knowledge
**Gesture Behavior Mapping:** Map each function to observed gestures
**Inferred Intention:** Explain overall communicative intent

[One-shot Example:]
**Input:** "I think this one is much better than the previous one." [Images]
**Output:** Motion Analysis: [Head, hands, arms, legs, body movements]
Function Derivation: [Comparison, Emphasis, Deixis functions]
Gesture Behavior Mapping: [Function-to-gesture relationships]
Inferred Intention: [Communication intent analysis]

[Data to be Annotated]

---

## C  ANNOTATION PROTOCOL AND VALIDATION

### C.1  MOTION PATTERN ANALYSIS

Unlike a action pattern Yang et al. (2025), the fine-grained movements for gestures is hard to be detect and analysis. To achieve this goal, we propose a rule-based algorithm for classifying temporal motion patterns by analyzing the geometric relationships between trajectory extrema and boundaries. Given a motion sequence $\mathbf{y} \in \mathbb{R}^T$ (e.g., joint angles or hand positions), our method extracts key statistics and determines the motion pattern through a deterministic decision process, as detailed in Algorithms 1–2.

The algorithm operates in three stages. First, it computes fundamental statistics: boundary values $(y_0, y_T)$, global extrema $(y_{\max}, y_{\min})$ with their indices, and the motion range $\delta = y_{\max} - y_{\min}$ (Algorithm 1, lines 3–6). If $\delta$ falls below a static threshold $\epsilon_s$, the motion is classified as linear/static, avoiding misclassification of noise as complex patterns (Algorithm 1, lines 8–10).

For non-static motion, the algorithm analyzes **extrema-boundary relations** by computing boolean indicators for whether the start/end positions are near (within tolerance $\epsilon$) the global extrema, and whether extrema occur in the trajectory interior (Algorithm 1, lines 12–14). These geometric relations capture motion characteristics invariant to scale and translation.

Finally, pattern classification applies hierarchical logical rules based on these relations (Algorithm 2). For instance, if the trajectory starts near one extreme and ends near the opposite

Table 8: **Test-time annotation prompt without visual grounding.** To prevent data leakage in evaluation, test-time annotations deliberately exclude visual information, requiring functions and intentions to be inferred solely from linguistic content.

---

**System Prompt:**
Assume you are the annotator for human speech. Without access to gesture images, you need to infer likely communicative functions and intentions from linguistic content alone. Based on Function Derivations, analyze the words and its durations within the transcript. Then analyze the Inferred Intention. The Definition of Function Derivation are as follows:

[Function Derivation: 16 classes of Function Derivations]

**User Prompt:**
I will provide you with:
- Previous two sentences for context
- Current sentence to be annotated
- *No visual information or keyframes*

Please analyze the linguistic content and provide predictions as follows:

**Linguistic Analysis:**
- Identify key words and phrases that typically trigger gestures
- Note speech elements that commonly correlate with specific movements
- Analyze the syntactic and semantic structure that implies gesture potential

**Function Derivation:** Infer likely functions based solely on linguistic content
**Predicted Gesture Types:** Suggest probable gesture categories without seeing actual movements
**Inferred Intention:** Predict the likely communicative intent based on linguistic cues

[One-shot Example for In-Context Learning without visual data]
[Data to be Annotated - transcript only]

---

$(s_{\max} \wedge e_{\min}) \vee (s_{\min} \wedge e_{\max})$, it's classified as a "round trip" pattern (Algorithm 2, lines 2–3). Other patterns include "return to extreme" (starting and ending at the same extreme), "peak between" (both extrema in the interior), and "single extreme inside" (one interior extreme), among others.

The algorithm employs context-aware thresholds that adapt based on motion type (e.g., different sensitivity for hand positions vs. joint angles) and achieves $\mathcal{O}(T)$ complexity through efficient single-pass operations (Algorithm 2). This deterministic approach provides interpretable pattern detection without requiring training data, making it suitable for real-time motion analysis applications where understanding the type of movement (cyclic, monotonic, or complex) is crucial for downstream tasks.

## C.2 TRAINING-TIME ANNOTATION PROTOCOL (WITH MOTION FRAMES)

VLMs have demonstrate their effectiveness in visual reasoning Yu et al. (2025). To construct training annotations, we leverage the this ability of VLMs and prompt GPT-4o-mini with both linguistic and visual inputs. Each prompt includes: **(1)** The two previous sentences spoken by the speaker, serving as linguistic context. **(2)** The current sentence to annotate, segmented into word units with corresponding timestamps. **(3)** The sampled starting and ending keyframe image for each word, together with the rule-based motion description annotation for the poses. We show the prompt template in Tab.7. The model is instructed to generate a structured analysis with the following outputs:

**Motion Analysis:** Detailed natural language description of body movements, including head motion, arm/shoulder gestures, finger positions, torso orientation, and stance.

**Function Derivation:** Identification of pragmatic functions (e.g., Emphasis, Deixis, Negation) that are linguistically relevant to the current sentence.

**Gesture Behavior Mapping:** Mapping between derived functions and observable gesture types (e.g., pointing, nodding, brow raise) following established gesture theory.

Table 9: **Baseline annotation prompt without structure.** This naive protocol excludes gesture theory or function derivation, asking the model to directly infer the speaker's communicative intent. This leads to overgeneralized or underspecified outputs.

---

**System Prompt:**
You are an assistant that helps interpret the meaning behind a speaker's body language and words. Given the speaker's sentence and gesture images for each word, describe what the speaker is trying to express overall. Do not break the task into components; simply provide an intention summary based on what you perceive.

[No prior gesture theory, no function derivation definitions]

**User Prompt:**
I will give you:
- A transcript of the speaker's sentence
- An image for each word the speaker says

Please describe what the speaker is trying to express or communicate. Use natural language, and focus on the overall message or feeling you perceive.

**Output:**
- One or two sentences summarizing the speaker's communicative intention
- Do not perform motion breakdown or gesture labeling
- Do not mention gesture function classes or mappings

[Example:]
**Input:** "I think this one is much better than the previous one." [Images]
**Output:** The speaker is expressing a strong preference for a current choice, likely implying confidence or satisfaction.

[Data to be Annotated]

---

**Inferred Intention:** A communicative goal inferred from the alignment of motion and function (e.g., emphasizing contrast, directing attention, expressing uncertainty).

This protocol captures visually grounded, multi-level annotation aligned with both motion and speech.

## C.3  TEST-TIME ANNOTATION PROTOCOL (TRANSCRIPT ONLY)

To avoid potential data leakage in test annotations, we exclude visual motion input from the VLM prompts during test set annotation. Each test prompt contains the two prior sentences for context and the current sentence to be annotated. No keyframes or motion descriptions are provided. The VLM is instructed to: **(1)** Infer likely communicative functions based solely on linguistic content. **(2)** Derive high-level communicative intent without visual grounding, as shown in Tab.8.

This simulates the actual evaluation scenario, where gesture models must predict motion solely from speech, and prevents the test set annotations from being conditioned on ground-truth poses.

## C.4  BASELINE ANNOTATION PROTOCOL (NO STRUCTURED PROMPT)

To examine the importance of structured reasoning, we design a baseline annotation protocol that omits the function derivation and gesture behavior mapping stages. In this setting, GPT-4o-mini is prompted with the current sentence and visual frames for each word, but is asked only to provide an inferred intention directly—without performing intermediate motion analysis or reasoning about communicative function. We present the prompt example in Tab.9.

This resembles a generic captioning-style instruction (e.g., "Describe what the speaker is trying to express"), lacking any prior definitions or decomposition of gesture semantics. While this setup may yield fluent outputs, it often results in: **(1) Overgeneralization:** Outputs tend to collapse nuanced signals (e.g., emphasis, negation, deixis) into vague descriptions such as "the speaker is sharing a thought." **(2) Hallucination:** In the absence of reasoning stages, the model may infer incorrect

Table 10: **Comparative annotation outputs across two utterances.** Structured annotations include function derivation and gesture mapping. Improper annotations suffer from overgeneralization, hallucination, or lack of compositionality.

| | |
|---|---|
| **Utterance A:** "I think watching anime is helpful for me" | |
| **Training-Time (w/ motion)** | **Function Derivation:** *Deixis* ("me"), *Mental State* (positive belief).
**Gesture Mapping:** Deixis → hand at chest, Mental State → relaxed stance.
**Inferred Intention:** The speaker reflects personally on the benefit of anime. Gestures reinforce introspection and confidence. |
| **Test-Time (transcript-only)** | **Function Derivation:** *Deixis*, *Mental State*.
**Gesture Mapping:** [Not available]
**Inferred Intention:** The speaker shares a personal viewpoint with implied conviction, likely supported by subtle gestures. |
| **Improper: Flat Intent Only** | **Inferred Intention:** The speaker is talking about anime.
*[Missing: No function derivation, no motion context, no gestural insight.]* |
| **Improper: Hallucinated Purpose** | **Inferred Intention:** The speaker is encouraging the audience to try watching anime as a productivity tool.
*[Issue: Adds persuasive intent not supported by transcript or body motion.]* |
| **Improper: Misaligned Gesture Mapping** | **Inferred Intention:** The speaker is contrasting anime with something unhelpful.
*[Issue: Misinterprets positive reflection as contrast/negation.]* |
| **Utterance B:** "I always try to move as much as I can when I'm not working" | |
| **Training-Time (w/ motion)** | **Function Derivation:** *Emphasis* ("working"), *Negation* ("not working"), *Modal* ("can").
**Gesture Mapping:** Emphasis → steady hands reinforce commitment; Negation → assertive fist posture; Modal → gestural space around "can".
**Inferred Intention:** Speaker emphasizes an active lifestyle outside of work. Gestures signal assertion and capability. |
| **Test-Time (transcript-only)** | **Function Derivation:** Same as above (*Emphasis*, *Negation*, *Modal*).
**Gesture Mapping:** [Omitted]
**Inferred Intention:** The speaker frames movement as a conscious, empowering action. Likely gestures reinforce contrast and agency. |
| **Improper: Flat Intent Only** | **Inferred Intention:** The speaker is saying that they move around a lot.
*[Issue: No deeper intent, no gesture mapping, missing compositional structure.]* |
| **Improper: Misaligned Functions** | **Inferred Intention:** The speaker is unsure whether they move enough and seems to compare working vs. resting.
*[Issue: Misses clear assertion and negation. Misreads modality.]* |
| **Improper: No Composition** | **Inferred Intention:** The speaker likes to be active.
*[Issue: Oversimplifies the sentence; collapses nuanced components (modal vs. negation vs. emphasis) into a flat label.]* |

intentions (e.g., persuasive intent where none exists). **(3) Loss of Interpretability:** Since outputs are not grounded in functional structure, they cannot be mapped to gesture execution in a controllable or compositional way. This baseline highlights the necessity of structured prompting in generating interpretable and semantically grounded gesture annotations. We include comparative examples in Tab. 10 to illustrate these failure modes in context.

## C.5   ANNOTATION VALIDATION AND HUMAN PREFERENCE STUDY

To assess the reliability of our annotation pipeline, we randomly sampled 100 utterances from the training set. Each sample was annotated using both the training protocol (with-motion) and the test protocol (transcript-only). Separately, expert annotators were provided with: **(1)** The utterance and its transcript. **(2)** The full sequence of rendered motion frames.

Experts then independently labeled: **(1)** The communicative function(s) present. **(2)** The inferred intention based on motion and speech. **(3)** The gesture types observed in the motion.

We then presented annotators with three candidate annotations for each sample (training VLM, test VLM, and human-generated), blinded and randomized. Annotators were asked to rate: **(1)** Which annotation most accurately reflected the speaker's intent. **(2)** Which annotation was most clearly and consistently reasoned.

Results, shown in main paper Fig.4, indicate that the training-style annotation (with visual grounding) achieved the highest human preference. However, the transcript-only test-style annotations also achieved strong scores, outperforming human-generated annotations in clarity and structural alignment. This validates the effectiveness of our prompt design and supports the use of VLM-generated labels for both training and evaluation.

C.6 VLM CONSISTENCY AND HALLUCINATION AUDIT

To ensure the reliability of our VLM-based annotation pipeline, we performed two targeted sanity checks: a consistency audit and a hallucination spot check.

**Consistency Under Repeated Prompts.** We randomly selected 100 utterances from the dataset and re-prompted GPT-4o-mini three times each under the same configuration. We examined the stability of the output across three categories: (i) function derivations, (ii) inferred intentions, and (iii) gesture behavior mappings. Across the 300 trials: 93% of the outputs maintained consistent function derivation labels. 84% preserved consistent gesture mappings across trials. These results suggest that the model exhibits stable behavior under repeated prompting, with low variance in the output of structural annotations.

**Hallucination Spot Check.** To assess the faithfulness of annotation outputs to visual evidence, we conducted an expert spot check on 50 randomly sampled annotation instances. Each instance included three components: **(1) Motion Descriptionz**, **(2) Function–Gesture Mapping**, and **(3) Inferred Intention**. For motion descriptions, 4 out of 50 samples (8%) were flagged for partial inconsistencies. These typically involved subtle over-interpretations—e.g., stating a "brow raise" when the face appeared neutral in the keyframe. No instances of fully fabricated or unrelated gestures were identified. For Function–Gesture Mapping, only 1 sample (2%) was marked as problematic, where a mapping relation (e.g., from a deictic phrase to a pointing gesture) was missing. The issue stemmed from under-specification rather than misalignment. For intention inference, 3 samples (6%) were flagged for slight exaggerations—such as over-interpreting neutral tones as emphasizing emotion. These were still broadly reasonable within the context of the utterance, and none were deemed to be outright hallucinations. Overall, the hallucination rate was low, and all identified issues were minor and recoverable. Importantly, no samples exhibited completely incorrect reasoning or disjointed alignment. This suggests the annotations are well-grounded and highlights the strong prompt-following and contextual inference abilities of the VLM. We also observe that minor hallucinations in motion description do not meaningfully degrade the accuracy of intention inference, supporting the robustness of our pipeline.

C.7 HUMAN STUDY INSTRUCTIONS

We present the details how we conducted the manual hallucination checking from the users as follows.

**Study 1: Function–Gesture Mapping Coherence  Objective:** Evaluate whether gestures are appropriate and coherent realizations of their corresponding communicative functions.

**Instructions to Annotators:** You are provided with a communicative function label (e.g., "Emphasis") and a corresponding gesture description (e.g., "Right hand performs rhythmic beat"). Please assess whether the described gesture appropriately fulfills or expresses the given function.

- Q1: Is this mapping coherent? (Yes / No)
- Q2 (Optional): If you selected "No", briefly explain why.

**Evaluation Protocol:** We randomly selected 50 samples and recruited 2 expert annotators. Final coherence score is computed as the average percentage of "Yes" responses across raters.

**Study 2: Motion Description–Keyframe Fidelity  Objective:** Determine whether the motion description accurately reflects the visible pose and dynamics presented in the keyframes.

**Instructions to Annotators:** You are shown a short video segment (or sequence of static keyframes) and a motion description (e.g., "Left hand slowly rises while the head turns right"). Please judge whether the described motion is clearly and accurately visible in the keyframes.

- Q1: Does the motion description match the keyframes? (Yes / Partially / No)
- Q2 (Optional): If "Partially" or "No", please explain which aspects were inaccurate or missing.

**Evaluation Protocol:** We used the same 50 annotated samples and had each rated by 2 human experts. Final scores are reported as the percentage of samples rated "Yes" (fully correct) and "Partially".

**Study 3: Inferred Intention Plausibility  Objective:** Assess whether the inferred communicative intention is a reasonable high-level summary of the utterance and accompanying gesture behavior.

**Instructions to Annotators:** You are shown a spoken utterance (text transcript) and a corresponding intention inference (e.g., "The speaker is attempting to reassure the listener about a concern"). Please judge whether the intention is plausible based on the content and tone of the utterance.

- Q1: Is the inferred intention plausible given the utterance? (Yes / Somewhat / No)
- Q2 (Optional): If "Somewhat" or "No", please describe why the inference may be overstated or misaligned.

**Evaluation Protocol:** Each of the 50 samples was evaluated by 2 annotators. We report the percentage of "Yes" and "Somewhat" responses to quantify plausibility and over-interpretation.

### C.8 QUALITATIVE COMPARISON: STRUCTURED VLM VS. FREE-FORM HUMAN ANNOTATIONS

To make the annotation protocols more concrete, we provide qualitative examples comparing our structured VLM annotations to the free-form human baseline described in Sec. 3.2. For each utterance, we show the transcript segment, a summary of the dominant motion pattern, the free-form human description, and our VLM-based motion–function–intention annotation. These examples illustrate that (i) both humans and the VLM operate over the same ontology and motion evidence, and (ii) our structured pipeline encourages more explicit function labels and gesture mappings, which are easier to use as supervision for gesture generation.

## D  IMPLEMENTATION DETAILS

**Hierarchical Audio-Motion Modality Alignment.** We adopt a dual-tower CLIP-based contrastive framework inspired by Tango Liu et al. (2024a), trained using a global InfoNCE loss. A key design choice for handling audio-motion modality alignment is the separation into low-level and high-level encoders.

For the audio stream, we represent input as raw waveforms and apply a 7-layer CNN (low-level) followed by a 3-layer Transformer (high-level), following the design of Wav2Vec2 (Baevski et al., 2020). For motion, we use a 15D representation and employ a 3-layer residual CNN (adapted from the Momask Motion Tokenizer (Guo et al., 2024)) and a 3-layer Transformer.

We use a projection MLP to process low-level features and another projection MLP with mean pooling for high-level features. Both audio and motion streams are temporally downsampled by a factor of 4.

**Local and Global Contrastive Loss.** We retain the InfoNCE loss over CLS tokens for global alignment, and additionally introduce a frame-level local contrastive loss. We treat frames within a temporal window $(i \pm t)$ as positives and distant frames $(i - kt, i - t, i + t, i + kt)$ as negatives, with $t = 4$ and $k = 4$ under a 30 FPS setting. This localized loss encourages robustness to minor temporal misalignments common in natural talking scenarios.

---

**Example 1**

---

**Transcript (snippet):**
"So *this part* is especially important for the final result."
**Motion summary:**
Right hand moves forward with a medium-amplitude beat on "this part", then retracts; slight head nod on "important".
**Free-form human annotation:**
"The speaker uses the right hand to emphasize 'this part' and nods to stress the importance of the point."
**Structured VLM annotation (ours):**
*Functions:* Emphasis; Mental State (importance).
*Gesture behavior:* Right-hand beat gesture toward the listener on "this part", followed by retraction; single head nod on "important".
*Inferred intention:* The speaker highlights a critical component of the explanation and signals that the audience should pay special attention to it.

---

**Example 2**

---

**Transcript (snippet):**
"On the one hand, we save time, and on the other, we reduce errors."
**Motion summary:**
Both hands alternate laterally: right hand extends to the right on "one hand", left hand extends to the left on "other". Mild torso sway follows the alternation.
**Free-form human annotation:**
"The speaker contrasts two aspects using both hands, pointing to each side while explaining the two options."
**Structured VLM annotation (ours):**
*Functions:* Contrast; Deixis.
*Gesture behavior:* Right hand extends to the right space on "one hand", left hand extends to the left space on "other", forming a lateral contrast between two alternatives. Torso subtly follows the side-to-side motion.
*Inferred intention:* The speaker frames the two benefits as parallel but distinct options, using spatial contrast to help the listener distinguish them.

---

**Example 3**

---

**Transcript (snippet):**
"I *think* this is probably not the best approach."
**Motion summary:**
Slight head tilt and brow raise on "think"; small outward palm-up gesture with both hands on "not the best".
**Free-form human annotation:**
"The speaker shows uncertainty or reservation, tilting the head and opening the palms while expressing doubt about the approach."
**Structured VLM annotation (ours):**
*Functions:* Mental State (epistemic stance); Attitude / Evaluation; Softened Negation.
*Gesture behavior:* Brief head tilt and eyebrow raise marking internal reflection on "think"; low-amplitude, palm-up gesture suggesting hesitation and mild disagreement on "not the best".
*Inferred intention:* The speaker carefully signals a skeptical evaluation while softening the disagreement, indicating personal opinion rather than categorical rejection.

---

Table 11: Examples comparing the free-form human baseline and our structured VLM annotations. Both use the same communicative function ontology and access to transcript + motion evidence, but the structured protocol encourages explicit function labels and gesture behavior descriptions that are easier to use as supervision for gesture generation.

**Stop-Gradient on Low-Level Encoders.** To jointly optimize both low- and high-level representations, we stop the gradient from the global InfoNCE loss to the low-level encoders, as in Tango Liu et al. (2024a). This design promotes feature learning across hierarchy levels.

**Intentional Gesture Tokenization.** We design the motion tokenizer using a simplified version of the encoder architecture above, followed by a decoder that mirrors its structure. To stabilize training, we reduce both to a single Transformer layer but maintain the same residual CNN blocks. The latent feature dimension is set to 512.

We apply a self-attention layer to project the 512-dimensional encoding to 32 dimension for quantization. The quantizer comprises 8 codebooks, with a dimension 32 and 8192 codes. For post-quantization, another attention layer maps the 32D features back to 512D for decoding.

**Intentional Gesture Generator.** The generator operates on token sequences produced by the tokenizer. It uses a Transformer with DiT Peebles & Xie (2023) architecture with 8 layers, a hidden dimension of 256, and a feedforward dimension of 1024, and number of head to be 4. In each layer, there is one self-attention, one cross-attention and followed with the feed-forward layer. For the cross-attention layer, due to two levels of audio conditioning, we design the structure of **Decoupled Cross-Attention.** Rather than forcing a single attention over mixed features, we apply two cross-attention branches separately. Given a shared query $Q$, we compute:

$$\mathcal{Z}_r = \text{SoftMax}\left(\frac{QK_r^\top}{\sqrt{d}} + \mathbf{P}\right) V_r, \quad \mathcal{Z}_i = \text{SoftMax}\left(\frac{QK_i^\top}{\sqrt{d}} + \mathbf{P}\right) V_i, \quad (4)$$

where $(K_r, V_r)$ and $(K_i, V_i)$ are key-value pairs from rhythmic and intentional features, respectively. The outputs $\mathcal{Z}_r$ and $\mathcal{Z}_i$ are summed to form the final conditioning representation.

This design introduces only a minimal overhead—adding separate key and value projections (only adding 2% parameters) for each cross-attention layer—yet yields consistent improvements of 0.01–0.03 in FGD across validation runs. This demonstrates the benefit of explicitly modeling disentangled prosodic and semantic cues during gesture generation.

**Optimizer Settings.** All modules are trained using the Adam optimizer Kingma (2014), with a learning rate of $1 \times 10^{-4}$, $\beta_1 = 0.5$, and $\beta_2 = 0.999$. We utilize a liner schedule with constant decay for the learning rate for the model learning. The generator is trained on 800 epochs for both single speaker setting and multi-speaker setting.

# E  ADDITIONAL EXPERIMENTS

**Baseline Methods.** We compare against a comprehensive set of recent gesture generation approaches Habibie et al. (2021); Liu et al. (2022a;b; 2023); Chen et al. (2024b); Yi et al. (2023); Liu et al. (2024b); Xu et al. (2024); Liu et al. (2025a), all evaluated under the **1-speaker setting** for fair comparison. This setting is used by most prior works and allows precise alignment with publicly reported results on BEAT-2.

Table 12: The quantitative results on BEAT-2. We bold the best results.

| Methods | FGD ($\downarrow$) | BC ($\rightarrow$) | Diversity ($\rightarrow$) |
|---|---|---|---|
| Ground-Truth | – | 0.703 | 11.97 |
| HA2G Liu et al. (2022c) | 1.232 | 0.677 | 8.626 |
| DisCo Liu et al. (2022a) | 0.942 | 0.643 | 9.912 |
| CaMN Liu et al. (2022b) | 0.664 | 0.676 | 10.86 |
| DiffSHEG Chen et al. (2024b) | 0.714 | 0.743 | 8.21 |
| TalkShow Yi et al. (2023) | 0.621 | 0.695 | 13.47 |
| ProbTalk Liu et al. (2024b) | 0.504 | 0.771 | 13.27 |
| EMAGE Liu et al. (2023) | 0.551 | 0.772 | 13.06 |
| Audio2PhotoReal Ng et al. (2024) | 1.02 | 0.550 | **12.47** |
| MambaTalk Xu et al. (2024) | 0.536 | 0.781 | 13.05 |
| SynTalker Chen et al. (2024a) | 0.469±0.13 | 0.736±0.04 | 12.43±0.23 |
| GestureLSM Liu et al. (2025a) | 0.409±0.03 | 0.714±0.12 | 13.24±0.23 |
| Intentional-Gesture | **0.379**±0.05 | **0.690**±0.04 | 11.00±0.21 |

**Full Generation Results.** Table 12 presents the quantitative results on the BEAT-2 benchmark. Our model, **Intentional-Gesture**, achieves state-of-the-art performance across all key metrics. Notably, our method obtains the lowest FGD (**0.379**), indicating the highest overall realism, while maintaining strong beat consistency (0.690) and natural motion diversity (11.00). These results demonstrate the benefit of our intentional alignment and conditioning mechanisms in generating gestures that are both semantically expressive and rhythmically precise.

**Results on Audio2PhotoReal.** Table 13 presents the quantitative results on the Audio2PhotoReal Ng et al. (2024) benchmark. Our model, **Intentional-Gesture**, achieves state-of-the-art performance across all key metrics. These results demonstrate the benefit of our intentional alignment and conditioning mechanisms in generating gestures can also be generalizable to dyadic conversational speaking and listening settings.

Table 13: The quantitative results on Audio2PhotoReal. We bold the best results.

| Methods | FGD ($\downarrow$) | Diversity ($\rightarrow$) |
|---|---|---|
| Ground-Truth | – | 2.50 |
| EMAGE Liu et al. (2023) | 4.43 | 2.13 |
| Audio2PhotoReal Ng et al. (2024) | 2.94 | 2.36 |
| GestureLSM Liu et al. (2025a) | 2.64 | 2.34 |
| Intentional-Gesture | **2.21** | **2.43** |

**Effect of Speaker Diversity on Retrieval.** To examine how speaker diversity influences model generalization, we fix the total number of training

Table 14: **Ablation on Speaker Diversity.** Increasing speaker diversity consistently boosts retrieval for both seen (*Known*) and unseen (*Unknown*) speakers, indicating better generalization.

| | Known | | | Unknown | | |
|---|---|---|---|---|---|---|
| Num | R@1↑ | R@5↑ | R@10↑ | R@1↑ | R@5↑ | R@10↑ |
| 1 | 20.63 | 40.34 | 50.67 | 1.03 | 1.95 | 2.56 |
| 2 | 29.41 | 57.63 | 60.61 | 1.44 | 2.31 | 2.78 |
| 3 | 31.37 | 60.42 | 63.39 | 1.67 | 2.49 | 2.92 |
| 4 | 33.52 | 63.52 | 66.87 | 1.87 | 2.64 | 3.01 |

samples and vary only the number of distinct speakers contributing data. As shown in Tab. 14 (right), increasing the number of training speakers from 1 to 4 significantly improves retrieval performance across both **in-domain** (seen speakers) and **out-of-domain** (unseen speakers) settings.

Notably, for in-domain cases, Recall@1 rises from 20.63% (1 speaker) to 33.52% (4 speakers), while for out-of-domain speakers, Recall@1 improves from 1.03% to 1.87%. These gains indicate that speaker diversity not only enriches the representation space but also enables more robust cross-speaker generalization. We hypothesis that training with a wider range of gestural patterns allows the model to better disentangle speaker-specific motion from shared semantic-rhythmic alignment.

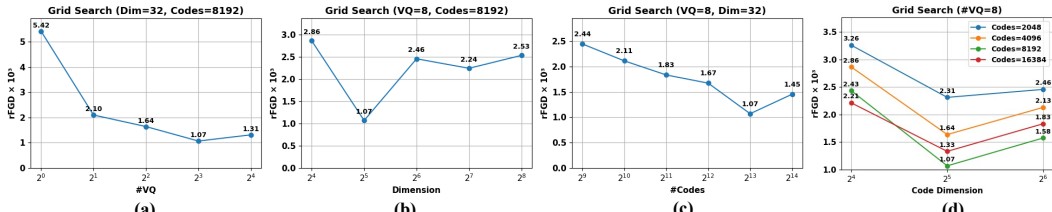

(a)  (b)  (c)  (d)

Figure 9: **Tokenizer ablation.** We perform both global and local grid searches to study the effects of codebook design choices on rFGD ($\times 10^3$). (a)–(c): Global sweeps varying one factor at a time; (d): Local grid search over code count and dimension. All results confirm consistent trends: 8 codebooks, a code dimension of 32, and 8192 codes yield optimal or near-optimal performance.

**Design Analysis.** We ablate design choices of the tokenizer, including the number of codebooks, code dimension, and code size. Fig. 9 shows that (1) 8 codebooks outperform fewer or more, balancing representational capacity and model compactness; (2) a code dimension of 32 achieves the best trade-off between expressiveness and compression; and (3) increasing code size improves rFGD up to 8192 codes, with diminishing returns beyond. These trends are consistent across global and local grid searches. For architecture design, we discover CNN presents better reconstruction quality, but the transformer presents better generation FGD. Our hybrid design takes the advantage of both variants.

**Long Sequence Generation Quality.** In the main paper, the experiment setting were conducted to generate sequences for the whole testing sequence. Specifically, we follow the existing works Liu et al. (2023; 2025a); Chen et al. (2024b) to utilize a sliding window for long sequence generation (with an average of 65.66 seconds). Each time, we provide the previous 2.13 seconds (a sequence length of 16 for neural representation) generated from the previous generated segment as the condition for the current time segment. Naturally, this setting is easy to encounter the error propagation issue (if the sequence from the previous generation present low quality, this error will be propagated to the current time segment). To understand this effect, we further design the new setting that replicate the inference setting of the same inference audio length as that utilized during training (8.633 seconds). We present the comparison setting between EMAGE, GestureLSM and Intentional-Gesture for single speaker setting in Tab.15. On long sequences, our model achieves the best performance (FGD = 0.379, BC = 0.690, Div. = 11.00). Under short-sequence inference, our FGD further improves by 0.133 (to 0.246), closely matching the improvements of 0.140 and 0.107 seen for EMAGE and GestureLSM, respectively—indicating a consistent FGD gap of  0.12 across models. Note that BC is not reported (–) for 8.633 s segments, due to the tricky implementation to select the precise audio segments from full ground-truth sequences with the generation segments. These results underscore the impact of error accumulation in sliding-window co-speech gesture generation and motivate future work on mitigating segment-wise propagation.

Table 15: Comparison of long-sequence (full test sequences) vs. short-sequence (8.633 s) inference on the single-speaker setting.

| | Long-seq Generation | | | Short-seq Generation | | |
|---|---|---|---|---|---|---|
| | FGD↓ | BC→ | Div.→ | FGD↓ | BC→ | Div.→ |
| GT | | 0.703 | 11.97 | | 0.703 | 11.97 |
| *Single-speaker* | | | | | | |
| EMAGE Liu et al. (2023) | 0.570 | 0.793 | 11.41 | 0.430 | - | 9.57 |
| GestureLSM Liu et al. (2025a) | 0.408 | 0.714 | 13.24 | 0.301 | - | **12.12** |
| Ours | **0.379** | **0.690** | **11.00** | **0.246** | - | 10.21 |

**Quantizer Comparisons Analysis** To isolate the influence of architecture on tokenizer performance, we standardized all encoder–decoder backbones to our CNN+Transformer design, which we found consistently outperforms alternatives across various quantizers. Specifically:

(1) EMAGE Liu et al. (2023) originally uses separate VQ quantizers for upper body, lower body, and hands. We replace its CNN encoders with our ResNet-style CNN blocks and normalize codebook embeddings rather than using raw outputs. We keep the original codebook size and dimensionality to demonstrate how reducing dimension and increasing code count affects performance.

(2) For RAG-Gesture Mughal et al. (2025), we re-implement their encoder and decoder based on Latent Motion Diffusion from MotionLCM codebase Dai et al. (2024). The comparions indicates for the continuous representation, it is hard to present the motion latent with a compressed latent mean prediction from VAE encoder to ensure it is synchronized with the audio for generation.

(3) For ProbTalk Liu et al. (2024b), we maintain their design of product quantization while improve the encoder and decoder with our design. This comparison indicates the product quantization, while present a latent codebook split, unlike our codebook design of separate latent motion representation, presents an inferior performance.

(4) For GestureLSM Liu et al. (2025a), we maintain the design of 6 layers of codebooks for each body region (upper, lower and hands), which leads to 18 codebook in total. While this multi-codebook approach achieves competitive reconstruction, its reliance on separate decoders for sequential region generation reduces efficiency and harms overall motion quality.

# F    USER STUDY DETAILS

For user study, we recruited 20 participants with good English proficiency. To conduct the user study, we randomly select videos from GestureLSM Liu et al. (2025a), EMAGE Liu et al. (2023), CAMN Liu et al. (2022b) and ours. Each user works on 8 videos. The users are not informed of the source of the video for fair evaluations. A visualization of the user study is shown in Fig 10.

**Subjective Evaluation Protocol and MOS Statistics** We briefly summarize the subjective evaluation protocol and provide full statistics for the Mean Opinion Scores (MOS) reported in Tab. 2.

**Scale and anchors.** Participants rate each video on a 5-point Likert scale for: (1) realness, (2) speech–gesture synchrony, and (3) smoothness. We adopt standard textual anchors: 1 = very poor (clearly unnatural / unacceptable), 3 = acceptable (plausible but with noticeable artifacts), and 5 = excellent (natural and highly convincing). MOS is computed as the average rating across all raters and all clips for a given method and criterion.

**Statistics and significance.** Table 16 reports MOS as mean $\pm$ standard deviation, together with 95% confidence intervals computed via bootstrapping over clips. We further conduct paired Wilcoxon signed-rank tests between our method and each baseline for each criterion. Across all three criteria, our method significantly outperforms GestureLSM, EMAGE, and CAMN ($p < 0.01$).

# G    METRIC DETAILS

**Fréchet Gesture Distance (FGD).** We adopt Fréchet Gesture Distance Yoon et al. (2020) to quantify the distributional similarity between real and generated gestures. Inspired by FID in image generation, FGD compares mean and covariance statistics of latent features extracted from a pretrained network:

## Subjective Evaluation of Video Generation Quality

Thank you for participating in the evaluation.

**Instructions**:

Please watch each gesture video and rate the videos based on Three evaluation metrics,
1. Realness: How real the gesture is
2. Synchronization: Whether the gesture is synchronized with the audio
3. Smoothness: Whether the gesture is smooth and natural
Please rate each video on a scale of 1 to 5, where 1 is the lowest and 5 is the highest

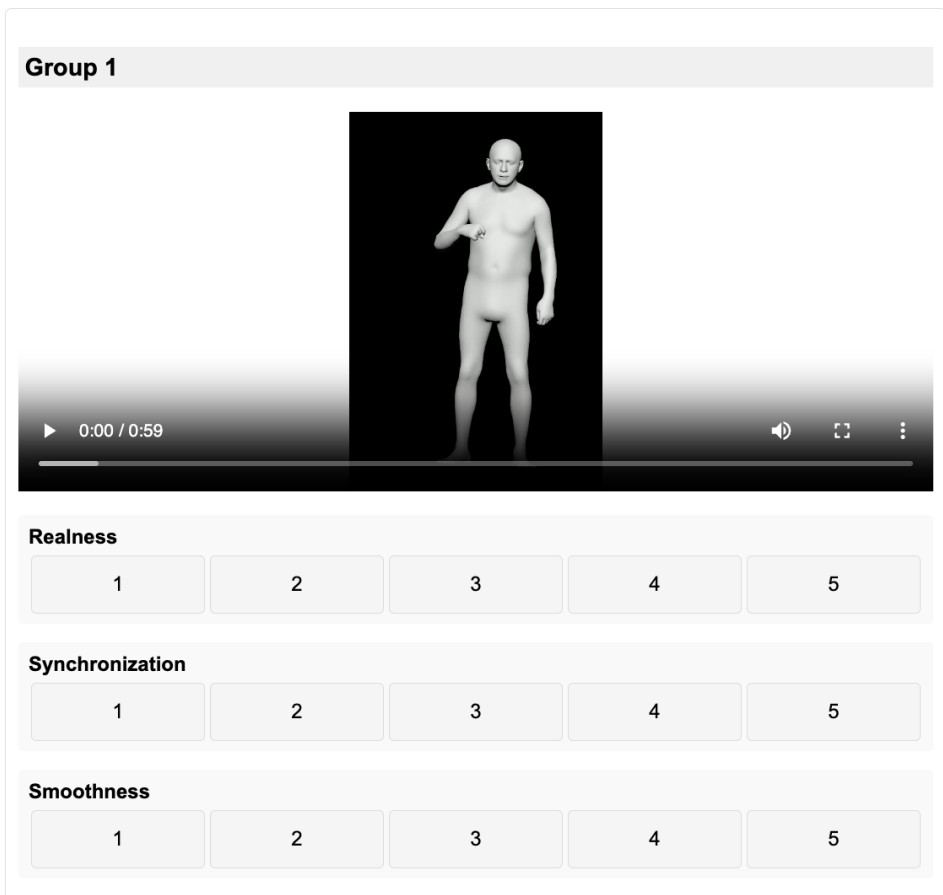

Figure 10: User Study Screenshot

| Method | Realness | Synchrony | Smoothness |
|---|---|---|---|
| CAMN | $1.34 \pm 0.12$ | $2.23 \pm 0.21$ | $2.14 \pm 1.23$ |
| EMAGE | $2.01 \pm 0.44$ | $2.42 \pm 0.13$ | $2.31 \pm 1.12$ |
| GestureLSM | $3.43 \pm 0.17$ | $3.61 \pm 0.22$ | $3.48 \pm 0.67$ |
| Ours | $3.76 \pm 0.21$ | $4.61 \pm 0.89$ | $3.92 \pm 0.32$ |

Table 16: MOS statistics (mean $\pm$ standard deviation) for each method and criterion.

$$\text{FGD} = \|\mu_r - \mu_g\|^2 + \text{Tr}\left(\Sigma_r + \Sigma_g - 2(\Sigma_r \Sigma_g)^{1/2}\right), \qquad (5)$$

where $(\mu_r, \Sigma_r)$ and $(\mu_g, \Sigma_g)$ are the empirical means and covariances of real and generated gesture embeddings, respectively. Lower FGD indicates better realism and distributional alignment.

**L1 Diversity (Div.).**    To assess sample-level variation, we compute L1 Diversity Li et al. (2021a), defined as the average pairwise L1 distance across $N$ generated sequences:

$$\text{L1 Diversity} = \frac{1}{2N(N-1)} \sum_{t=1}^{N} \sum_{j=1}^{N} \left\| p_t^i - \hat{p}_t^j \right\|_1 , \tag{6}$$

where $p_t^i$ and $\hat{p}_t^j$ denote the joint positions at frame $t$ for the $i$-th and $j$-th sequences. To focus on local articulation, global translation is removed before computing distances.

**Beat Constancy (BC).**    Beat Constancy Li et al. (2021b) measures rhythmic alignment between gesture dynamics and speech. Motion beats are detected as local minima in upper body joint velocity, while speech onsets define audio beats. BC is computed as:

$$\text{BC} = \frac{1}{|g|} \sum_{b_g \in g} \exp \left( -\frac{\min_{b_a \in a} \|b_g - b_a\|^2}{2\sigma^2} \right) , \tag{7}$$

where $g$ and $a$ are the sets of gesture and audio beats, respectively. BC closer to ground-truth implies stronger gesture-speech synchronization.

## H    ETHICAL STATEMENT

While our work is centered on generating human motion videos, it raises ethical concerns due to its potential misuse for photorealistic human motion retargeting. We emphasize the importance of responsible use and recommend implementing practices such as watermarking and deepfake detection to mitigate the risks involving deepfake videos and animated representations.

## I    REPRODUCIBILITY STATEMENT

We have provided the code of algorithmic annotation for the motion pattern analysis in the supplementary material together with the code for the whole system.

## J    THE USE OF LARGE LANGUAGE MODELS

We utilize Large Langauge Models for the dataset annotation and paper polishing.

## K    LIMITATIONS

While our framework demonstrates strong performance across alignment, tokenization, and gesture generation, several limitations remain.

First, our method relies on pre-annotated sentence-level intention descriptions to guide semantic learning. This setup assumes that such annotations are either available or can be reliably extracted, which may not hold in less curated or low-resource scenarios. Future work could explore unsupervised or weakly supervised intention discovery to broaden applicability.

Second, while the multi-codebook tokenizer introduces structure into the latent space, it does not guarantee complete disentanglement between semantic and rhythmic dimensions. Investigating more principled inductive biases or factorized token learning may improve interpretability and controllability.

Third, as shown in Sec.E, we discover that existing methods present error propagation issues for long-sequence generation settings. We would like to highlight this issue and hope future works can propose solutions for this fundamental issue for the co-speech gesture generation domain.

Fourth, in this work, while the motion description annotation, gesture-behavior function mapping are intermediate outputs during the annotation procedure, they are not input as variables for the

motion control but only intention annotations were utilized. We build this simple baseline because during inference procedure, we are not able to obtain these motion relationed analysis. However, we argue that the values of these annotations should not be ignore and hope future works can further explore the use cases of these annotations as well for motion control and inspire the analysis of the relationships between gesture motion patterns and linguistic cues from speech context.

Finally, although our hierarchical alignment improves generalization across speakers, domain shifts—such as significant accent variation, disfluency, or cultural gesture norms—remain challenging. Incorporating domain adaptation techniques or cross-cultural gesture modeling could enhance robustness in real-world deployments.

