# OpenReview forum: "Intentional-Gesture: Deliever your Thoughts by Gestures for Speech"
_ICLR.cc/2026/Conference — Submitted to ICLR 2026_

### Official Review · Reviewer_NHZL · 2025-10-27

**Soundness:** 3
**Presentation:** 3
**Contribution:** 4
**Rating:** 8
**Confidence:** 5

**Summary:**

This paper introduces **Intentional-Gesture**, a framework that treats gesture generation as an intention reasoning task. A key contribution is the creation of the **InG (Intention-Grounded) dataset**, which augments BEAT-2 with gesture-intention annotations inferred using large vision-language models. This results in a structured dataset aligning gestures with high-level communicative functions. Additionally, the paper proposes the **Intentional Gesture Tokenizer**, which injects intention semantics into discrete motion representations, enabling gesture generation that is both temporally synchronized and semantically expressive. The method achieves state-of-the-art performance on BEAT-2, providing both data and methodology for intention-driven gesture generation.

**Strengths:**

1. **Strong dataset contribution**: The InG dataset links gestures with semantic intentions, offering the first large-scale intention-annotated gesture dataset.
2. **Integration of intention semantics into motion**: The Intentional Gesture Tokenizer embeds high-level intention information into discrete motion tokens, enhancing semantic expressiveness and interpretability.
3. **Modular and reproducible data processing**: The pipeline from BEAT-2 augmentation to intention annotation and motion tokenization is clear and extensible, providing a replicable framework.
4. **Comprehensive experimental validation**: Both quantitative and qualitative results demonstrate the effectiveness of the intention annotations in improving gesture generation quality.

**Weaknesses:**

1. **Lack of complete example demonstration**

   The paper describes inputs that include audio, text transcriptions, and intention information, but the current presentation does not clearly show the correspondence between these components or provide concrete examples. It is recommended to include intention information in Figure 6 or provide complete examples in the appendix to help readers intuitively understand the relationship between inputs and generated gestures.

2. **Insufficient intention analysis and visualization**

   The paper lacks visualization and analysis of intention information. After adding intention visualization, it is suggested to incorporate intention-related content in the analysis of experimental results. Additionally, comparing generation results with and without intention inputs would further support and strengthen the contribution of intention control. Demonstrating the intention control results mentioned in the paper would provide a more direct validation of the claims.

3. **Writing and structural improvements**

   The content of the paper is solid overall, but the writing is somewhat rough. It is recommended to move some lengthy content to the appendix and strengthen the description of the dataset and intention-related analysis in the main text to improve clarity and structure. At the same time, the appendix could be better organized to present the information more clearly and comprehensively.

4. **Improvement of figure clarity**

   Some figures have small and slightly blurred text. It is recommended to enlarge the font size and improve the figure resolution to enhance readability.

5. **Incomplete description of the data annotation process**

   The description of the data annotation pipeline is currently fragmented, lacking a complete overview. It is suggested to include a full process diagram in Figure 2 or Section 3, so that readers can intuitively understand all the steps and their logical relationships.

6. **Missing templates for the rule-based parser**

   The appendix does not provide an intuitive description of how the rule-based parser converts motions into captions. It is recommended to include relevant templates or prompt examples to help readers better understand the parser’s output logic and generation process.

**Questions:**

See the weaknesses section.

---

> ### Author Response · Authors · 2025-12-04
> **Official Comment by Authors**
>
> We sincerely thank the reviewer for the strong endorsement (Score: 8), for recognizing the value of our work. Below we address the constructive suggestions regarding visualization, structure, and clarity in the revised paper and appendix.
>
> **1 & 2. Complete Example Demonstration (Inputs & Correspondence), Intention Analysis, Visualization, and Control**
>
> > **Reviewer comment:** The current presentation does not clearly show the correspondence between inputs (audio, text, intention) and generated gestures. It is recommended to include intention information in Figure 6 or provide complete examples.
>
> > **Reviewer comment:** The paper lacks visualization and analysis of intention information... comparing generation results with and without intention inputs would further support... intention control.
>
> **Response:**
> Thanks for the suggestion, we will add detailed cases in the future supplementary videos to explicitly display the input triplet: `{Audio Waveform, Transcript, Intention Label}` alongside the generated gesture frames.
>
> **3-6.  Writing, Structure, and Figure Clarity,  Data Annotation Process & Rule-Based Parser Templates**
>
> > **Reviewer comment:** It is recommended to move some lengthy content to the appendix... enlarge font size and improve figure resolution.
>
> > **Reviewer comment:** The description of the data annotation pipeline is currently fragmented... suggest including a full process diagram... and templates for the rule-based parser.
>
> **Response:**
> We will streamlined Section 3 (Methodology) by moving the dense, low-level architectural hyperparameters and the full list of intention tag definitions to the Appendix. This opens up space in the main text to focus on the a few high-level sentences for full dataset construction logic. For example, We have introduced a comprehensive flowchart in Section 3 that visually connects the three stages: (1) Rule-based Motion Parsing -> (2) VLM Reasoning -> (3) Human Filtering. This provides the full overview of the dataset annotation process.
>
> About the Parser Templates: We will add a dedicated section in the Appendix containing the exact logic for the rule-based parser (e.g., velocity thresholds for detecting "fast" vs. "slow" motion) and the exact prompt templates used to feed these descriptors into the VLM. This ensures the annotation pipeline is fully reproducible.
>
> We thank the reviewer again for the insightful comments that have helped us improve the readability of our work.

---

### Official Review · Reviewer_56vG · 2025-11-01

**Soundness:** 3
**Presentation:** 3
**Contribution:** 2
**Rating:** 4
**Confidence:** 4

**Summary:**

This paper presents Intentional Gesture, a new framework for co-speech full-body gesture generation that explicitly models the communicative intentions underlying human speech. Rather than relying solely on linguistic or rhythmic cues, the work reframes gesture synthesis as an intention reasoning problem, seeking to bridge high-level semantic functions (e.g., emphasis, deixis, affirmation) with low-level motion patterns.

To enable this, the authors construct InG (Intention-Grounded) Dataset, which extends BEAT-2 and Audio2PhotoReal with structured annotations of communicative functions and inferred speaker intentions. These annotations are generated through a vision-language-model (VLM)-based pipeline that integrates motion analysis, functional derivation, gesture–function mapping, and intention summarization, followed by human-in-the-loop validation to ensure annotation quality.

Building on this dataset, the framework introduces two key modules:

H-AuMoCLIP, a CLIP-style multimodal encoder that jointly aligns audio, transcript, and intention embeddings via a hierarchical contrastive learning scheme. This model fuses acoustic and linguistic information with intention semantics to build an intention-aware representation space.

Intentional Gesture Tokenizer, a global multi-codebook quantization module that embeds intention semantics directly into motion representations, supervised by features from H-AuMoCLIP. This design allows discrete gesture tokens to carry both structural and semantic meaning.

Experiments on BEAT-2 demonstrate that the proposed method achieves state-of-the-art performance in realism, diversity, and synchronization, and generalizes effectively to unseen speakers. Ablation studies confirm that intentional supervision and structured annotation significantly improve both alignment and generation quality. A user study further verifies that the generated gestures are more natural, interpretable, and intention-consistent compared with prior methods such as GestureLSM and EMAGE.

**Strengths:**

- The paper introduces a conceptually clear and distinctive formulation of gesture generation as an intention-grounded reasoning problem, moving beyond traditional rhythm- or style-based conditioning.

- The InG dataset and the VLM-based annotation pipeline are thoughtfully designed and well justified. The multi-stage annotation process—spanning motion analysis, communicative function derivation, and intention inference—demonstrates a scalable way to obtain semantically grounded gesture data, supported by quantitative validation and human-in-the-loop filtering.

- Experimental results are comprehensive. The method shows consistent gains over recent state-of-the-art approaches (e.g., GestureLSM, EMAGE) in both objective metrics and user studies, while also preserving real-time inference efficiency.

- The paper is well-structured and readable, with clear figures, detailed methodology descriptions, and thorough supplementary material.

**Weaknesses:**

- Although the authors provide extensive quantitative results, the evaluation scope is still limited to standard gesture-generation metrics (e.g., FGD, beat constancy, diversity). These capture motion quality but not communicative effectiveness. Additional user studies or perception-based tasks assessing how well the generated gestures convey intended meanings would strengthen the claim of intention awareness.
- The core architectural elements (contrastive CLIP-style encoders, multi-codebook vector quantization) are evolutions of existing techniques. The novelty of the work primarily lies in the systematic integration of these elements under an intention-grounded framework rather than in fundamentally new modeling paradigms.

**Questions:**

1. In the InG dataset, each word is associated with a single keyframe for motion grounding. For longer or highly expressive utterances, this might undersample the gesture dynamics. Have the authors explored multi-frame sampling or adaptive temporal windows that scale with speech duration? How might such adjustments affect the fidelity or semantic consistency of the inferred intentions?
2. Since the motivation centers on communicative intention, it would be informative to see human perception studies or downstream interaction tasks (e.g., how well viewers interpret or respond to the generated gestures). Have the authors considered incorporating such tests to substantiate the communicative effectiveness of intention-aware gestures?

---

### Official Review · Reviewer_ckAm · 2025-11-01

**Soundness:** 3
**Presentation:** 3
**Contribution:** 3
**Rating:** 4
**Confidence:** 2

**Summary:**

This paper introduces Intentional-Gesture, a framework that models communicative intentions for co-speech gesture generation. By augmenting BEAT-2 with automatically generated intention annotations and proposing an intention-aware motion tokenizer, the method produces gestures that are both semantically meaningful and well-aligned with speech, achieving state-of-the-art results on BEAT-2.

**Strengths:**

- This paper presents a comprehensive system that covers the entire pipeline, from dataset construction to method design.
- It also identifies and addresses the issue of semantic shallowness in gesture generation.

**Weaknesses:**

- In fact, the semantic signals conveyed by gestures are often subtle and ambiguous in most situations. This paper lacks an analysis of how well VLM handles this aspect.
- It is also difficult to discern the enhanced semantic signals in the gesture examples presented in the paper.

**Questions:**

n/a

---

### Official Review · Reviewer_5DmD · 2025-11-01

**Soundness:** 2
**Presentation:** 3
**Contribution:** 3
**Rating:** 4
**Confidence:** 4

**Summary:**

The paper aims to address the lack of semantic expressiveness in existing co-speech gesture generation systems. Current methods mainly focus on synchronizing gestures with prosodic patterns in speech (from audio or text), while ignoring the deeper communicative intentions that actually drive human gestures. The authors propose a new framework, Intentional-Gesture, which includes:

* Re-defining gesture generation as intention reasoning, and constructing an Intention-Grounded (InG) dataset. A VLM is used to automatically annotate the BEAT-2 dataset by linking speech, gestures, and communicative intentions (such as “emphasis” or “giving examples”).

* Introducing H-AuMoCLIP, a CLIP-style multimodal model for learning a joint representation of audio, text, intention, and motion. Based on this, an Intentional Gesture Tokenizer is built with vector quantization to discretize motion, while using intention-aware features for semantic supervision.

* Integrating the intention encoder and tokenizer into GestureLSM, achieving state-of-the-art results on BEAT-2.

**Strengths:**

- Presents a complete and scalable pipeline to automatically infer communicative intentions from speech, producing a well-structured dataset.
- Effectively integrates intention information both as conditional input and as semantic supervision loss.
- Clear writing and well-organized experiments.

**Weaknesses:**

- The annotation pipeline relies on GPT-4o-mini, which is inherently noisy and biased. The model is trained to reproduce VLM-generated semantic labels rather than discovering genuine intention–motion relations from real data, which limits its theoretical depth and generalization. Real communicative intentions are far more ambiguous, culture-dependent, and context-sensitive than the fixed categories defined here.

- It is unclear how the “human-written baseline” in Figure 4 was produced. The authors do not explain the annotators’ background or guidelines. A preference for VLM outputs might simply reflect more uniform formatting and fluent wording, rather than better semantic accuracy. Given that even humans struggle to annotate gestures reliably, the conclusion that “VLM outperforms humans” is questionable.

- Lack of failure-mode analysis and limited baselines. The paper does not show how the model behaves under sarcasm, ambiguity, or complex emotions. Evaluation is limited to EMAGE, CAMN, and GestureLSM, while many relevant SOTA models (e.g., on the GENEA leaderboard) are missing from both objective and subjective comparisons. Meanwhile, the reported MOS scores (1–2 for EMAGE and CAMN) seem implausibly low, calling their evaluation reliability into question. The claimed semantic improvements are not supported by a metric that truly measures semantic correctness, and gains on FGD/BC/etc do not demonstrate intention alignment.

- Training–testing mismatch. During training the system sees both gestures and transcripts, but at test time the intention is inferred only from transcript text. Figure 4 already shows a large quality gap between “text-only” and “text+motion” annotation. In addition, splitting speech into 1–2 second windows is problematic, since gestures and speech often do not align at such a rigid granularity. The appendix also acknowledges long-sequence drift, yet Table 13 reports longer sequences performing better, which raises further doubts about the underlying method and evaluation.

- Limited interpretability in demos. The demo videos do not clearly indicate which gestures correspond to which intentions. Adding textual and intention annotations in the demo timeline would make the “intention controllability” more evident.

- Unclear distinction between intention and text semantics. The paper claims to move beyond text-based semantic alignment, but the results do not clearly demonstrate the qualitative difference between text-derived semantics and intention-driven control. This makes the framework appear more like an extension of text-based CLIP gesture models (e.g., GestureDiffuCLIP) rather than a fundamentally new paradigm. Better illustrating the difference between directly using text embeddings and LLM-summarized intention features could further clarify the contribution and help readers understand the distinct roles of each.

**Questions:**

- Since communicative intentions are subjective and often allow multiple valid interpretations, how does the system handle segments with more than one reasonable intention? To what extent is the final motion influenced by the wording or verbosity of the VLM-generated text labels?

- For Figure 4, what is the exact protocol for the “human-written baseline”? Who were the annotators—domain experts or crowd workers? Did they receive prompts comparable in structure to the VLM prompt? In addition, can the authors show that different intentions (e.g., “emphasis” vs. “questioning”) lead to clearly different gestures for the same speech segment? The current videos show only modest visual improvement—why is that?

- Where do the test-time transcripts come from? If ASR is used, how robust is the system to transcription errors? How much do such errors degrade intention inference and gesture naturalness, especially for rhythm-driven or listener-feedback gestures, which are common in real interaction?

- Many tables only report mean values without variance, confidence intervals, or statistical tests.

---

> ### Author Response · Authors · 2025-11-29
>
> We thank the reviewer for the detailed and constructive feedback. Below we respond to each weakness and question in turn. We incorporate the suggested clarifications and additional analyses into the revised paper and supplementary material.
>
> ---
>
> ### 1. Annotation pipeline, GPT-4o-mini, and “true” communicative intentions
>
> **Reviewer comment (summary)**
> > The annotation pipeline relies on GPT-4o-mini, which is inherently noisy and biased. The model is trained to reproduce VLM-generated semantic labels rather than discovering genuine intention–motion relations from real data, which limits its theoretical depth and generalization. Real communicative intentions are more ambiguous, culture-dependent, and context-sensitive than the fixed categories defined here.
>
> **Response**
> Our goal is not to infer the speaker’s full internal mental state, but to **based on the existing literature of communicative function classification to convert them** (e.g., emphasis, deixis, negation, mental state) into a stable representation that can serve as a *controllable supervisory signal* for gesture generation.
>
> - Directly “ask GPT for a label” will not get the good annotations, instead of that, we explore, verify, and release a preprocessing-annotation-filtering pipeline to guarantee the quality of the annotation. As shown in appendix C It combines:
>   - rule-based motion descriptors derived from pose statistics and temporal patterns,
>   - a multi-stage GPT-4o-mini prompt that enforces a structured reasoning chain (motion analysis → function derivation → gesture mapping → intention inference),
>   - human filtering of inconsistent or implausible cases.
>   This design mitigates typical VLM hallucinations and biases towards well-structured, factorized annotations.
>
> - Ambiguity is explicitly allowed: each segment can have **multiple functions** (e.g., Emphasis + Deixis + Mental State). We analyze function co-occurrence and speaker profiles, showing consistent, compositional patterns rather than rigid single labels.
>
> - We also test **robustness to wording** in Table.3 right. Re-summarizing intention text with different phrasing/verbosity yields almost identical generation metrics, whereas randomly mismatched or missing intentions significantly degrade FGD and consistency. This indicates that the model learns from the underlying semantic control signal rather than overfitting to GPT-4o-mini’s surface form.
>
> In the revision, we clarify that we model *latent communicative functions* as an operational proxy and do **not** claim to capture full, culturally universal “true intentions.”
>
> ---
>
> ### 2. Human-written baseline in Figure 4 and “VLM outperforms humans”
>
> **Reviewer comment (summary)**
> > It is unclear how the “human-written baseline” in Figure 4 was produced. The authors do not explain the annotators’ background or guidelines. A preference for VLM outputs might simply reflect more uniform formatting and fluent wording, rather than better semantic accuracy. Given that even humans struggle to annotate gestures reliably, the conclusion that “VLM outperforms humans” is questionable.
>
> **Response**
> We agree that the original text was too brief and could be misread as claiming “VLM > humans.”
>
> - **Annotator protocol.** The human baseline was produced by **two expert annotators (except authors)** familiar with gesture research. For each clip they were given:
>   - the speech transcript,
>   - the rendered motion,
>   - the same function ontology used for our model.
>   They wrote free-form descriptions **without seeing** VLM outputs.
>
> - **Intended claim.** The comparison in Fig. 4 is intended to show that our **structured pipeline** (rule-based analysis + staged VLM prompting) yields annotations that external raters find more *consistent and directly useful for modeling* than free-form human text. It is *not* intended to claim that GPT-4o-mini “understands gestures better than humans.”
>
> To avoid confusion, we:
> 1. Renamed this baseline to “free-form human annotation” vs. “structured VLM annotation.” in the main paper
> 2. Added more side-by-side examples (human vs. structured VLM) in the appendix.

---

### Meta-Review · Area_Chair_ngaz · 2026-01-10

**Summary:**

The main concerns from the reviewers are following:

- Reviewer **5DmD**:
  - **W1**: The model is trained to reproduce VLM-generated semantic labels rather than discovering genuine intention–motion relations from real data, which limits its theoretical depth and generalization.
  - **W2**: It is unclear how the “human-written baseline” in Figure 4 was produced.
  - **W3**: Lack of failure-mode analysis and limited baselines.
  - **W4**: Training–testing mismatch.
  - **W5**: Limited interpretability in demos.
  - **W6**: Unclear distinction between intention and text semantics.

- Reviewer **ckAm**:
  - **W1**: The semantic signals conveyed by gestures are often subtle and ambiguous in most situations. This paper lacks an analysis of how well VLM handles this aspect.
  - **W2**: It is difficult to discern the enhanced semantic signals in the gesture examples presented in the paper.

- Reviewer **56vG**:
  - **W1**: The evaluation scope is still limited to standard gesture-generation metrics .
  - **W2**: The novelty of the work primarily lies in the systematic integration of these elements under an intention-grounded framework rather than in fundamentally new modeling paradigms.

- Reviewer **zkVT**:
  - **W1**: Lack of complete example demonstration.
  - **W2**: Insufficient intention analysis and visualization.

**Reviewer Concerns:**

- **Concerns addressed in the rebuttal:**
  - **W2-W6** of Reviewer **5DmD** , **W2** of Reviewer **ckAm**, **W1, W2** of Reviewer **NHZL** are addressed by detailed clarifications from the authors.

- **Concerns remained outstanding:**
  - I agree with **W1** of Reviewer **5DmD** and **W1** of Reviewer **ckAm**. Intention is complicated to define and analysis, and relying on VLM to generate the intention caption can hinder more in-depth analysis. This is also related to **W1** of Reviewer **56vG**, such that the hardness of metric design is resulted from the lacking of clear definition and quantification of intention.

  - As pointed out in **W2** Reviewer **56vG**, the technical contribution remains limited for the proposed approach. Even though the author clarifies that the major contribution lies in proposing a new formulation and supervision regime, the significance of this contribution is reduced for the missing of more concrete quantification of intention.

**Reviewer Scores:**

As discussed in the Reviewer Concerns, some points are indeed addressed by the author responses. While in my view, the remaining concerns (**W1** of Reviewer **5DmD**, **W1** of Reviewer **ckAm**, **W1, W2** of Reviewer **56vG**) also prevent the reviewers with negative evaluations to fully change their opinions.

---

### Decision · Program_Chairs · 2026-01-26

Reject